Journal of Data-centric Machine Learning Research (2025)       Submitted 01/25; Revised 07/25; Published 11/25

# Towards Human-Guided, Data-Centric LLM Co-Pilots

**Evgeny S Saveliev**[*]                                                        ES583@CAM.AC.UK
*University of Cambridge, Cambridge, UK*

**Jiashuo Liu**[*†]                                                        LIUJIASHUO77@GMAIL.COM
*Tsinghua University, Beijing, China*

**Nabeel Seedat**[*]                                                        NS741@CAM.AC.UK
*University of Cambridge, Cambridge, UK*

**Anders Boyd**                                                        A.C.BOYD@AMSTERDAMUMC.NL
*Amsterdam University Medical Centers, Amsterdam, NL*

**Mihaela van der Schaar**                                                        MV472@CAM.AC.UK
*University of Cambridge, Cambridge, UK*

**Reviewed on OpenReview:** *https: // openreview. net/ forum? id= MWOrjmelCI*

**Editor:** Matthias Feurer

## Abstract

Machine learning (ML) has the potential to revolutionize various domains and industries, but its adoption is often hindered by the disconnect between the needs of domain experts and translating these needs into robust and valid ML tools. Despite recent advances in LLM-based co-pilots to democratize ML for non-technical domain experts, these systems remain predominantly focused on model-centric aspects while overlooking critical data-centric challenges. This limitation is problematic in complex real-world settings where raw data often contains complex issues, such as missing values, label noise, and domain-specific nuances requiring tailored handling. To address this we introduce CliMB-DC, a human-guided, data-centric framework for LLM co-pilots that combines advanced data-centric tools with LLM-driven reasoning to enable robust, context-aware data processing. At its core, CliMB-DC introduces a novel, multi-agent reasoning system that combines a strategic coordinator for dynamic planning and adaptation with a specialized worker agent for precise execution. Domain expertise is then systematically incorporated to guide the reasoning process using a human-in-the-loop approach. To guide development, we formalize a taxonomy of key data-centric challenges that co-pilots must address. Thereafter, to address the dimensions of the taxonomy, we integrate state-of-the-art data-centric tools into an extensible, open-source architecture, facilitating the addition of new tools from the research community. Empirically, using real-world healthcare datasets we demonstrate CliMB-DC's ability to transform uncurated datasets into ML-ready formats, significantly outperforming existing co-pilot baselines for handling data-centric challenges. CliMB-DC promises to empower domain experts from diverse domains — healthcare, finance, social sciences and more — to actively participate in driving real-world impact using ML. CliMB-DC is open-sourced at: https://github.com/vanderschaarlab/climb/tree/climb-dc-canonical

**Keywords:**   Agent, Co-Pilot, Data-centric AI, Large Language Model

---

*. Equal Contribution

†. Research conducted while visiting the van der Schaar Lab, University of Cambridge.

## 1 Introduction

Over the past decade, machine learning (ML) has evolved at a breathtaking pace, raising hopes that advanced ML methods can transform a wide range of domains and industries. However, for many domain experts — including medical researchers, social scientists, business analysts, environmental scientists, education researchers and more — conceiving a problem through which ML can provide a solution remains challenging. Despite having a deep understanding of their data and domain-specific challenges, these individuals often lack the programming or technical background needed to implement sophisticated ML pipelines (Pfisterer et al., 2019), and thus are considered *non-technical domain experts*. This gap in expertise creates a significant barrier to realizing the potential of ML across these domains.

Recent advancements in large language models (LLMs) have paved the way for AI co-pilots that promise to automate various aspects of ML development through natural language interaction (Hassan et al., 2023; Tu et al., 2024). However, current co-pilots remain predominantly focused on model-centric aspects—such as architecture selection and hyperparameter tuning—while overlooking the fundamental role of the data-centric side to ML. Since data-centric aspects largely determine the performance, fairness, robustness and safety of ML systems, ignoring the processes of constructing and handling data can negatively affect performance or worse lead to incorrect conclusions. Unfortunately, real-world data often contains missing values, inconsistencies, mislabeled records, and domain-specific nuances (see Table 1) and thus the data is usually not ML-ready (Sambasivan et al., 2021; Balagopalan et al., 2024). Furthermore, applying a "one-size-fits-all" data cleaning script from an LLM co-pilot that cannot be tailored to the varying structures of data risks erasing critical signals or introducing biases, and leaves the domain experts powerless to intervene.

There is indeed a growing interest in data-centric AI within the ML community — emphasizing the importance of ML to improve data quality, curation, and characterization (Seedat et al., 2024; Zha et al., 2023; Liang et al., 2022). In particular, numerous data-centric ML tools and methods have been developed for handling common data issues, such as missing values, noisy labels, and data drift (Northcutt et al., 2021a; Jarrett et al., 2022; Seedat et al., 2022a, 2023b; Liu et al., 2023). However, for non-technical domain experts, these tools are often abstract to implement and remain out of reach to use. Integrating these tools into LLM-based co-pilots would not only allow tailored handling of data and thus empower domain experts, but also would broaden the use of data-centric AI research across various disciplines and application settings — including healthcare, finance, environmental studies, education, etc.

Despite their value, data-centric tools are not a panacea in and of themselves and cannot be applied by co-pilots in isolation. Actions like imputing data or rectifying noisy labels require contextual understanding to avoid distorting critical domain-specific information. This underscores the need for expert oversight—guidance from individuals deeply familiar with the nuances of the data—to ensure that actions align with domain-specific goals and constraints. Such guidance is crucial in high-stakes fields like healthcare and finance, where improper data handling can lead to misleading conclusions or harmful decisions.

This interplay between human expertise and data-centric automation presents a unique challenge for LLM-based co-pilots. Designing systems capable of nuanced reasoning and iterative planning, while effectively incorporating expert feedback, remains a significant

hurdle. A co-pilot must not only execute tasks but also intelligently sequence and adapt data processing pipelines with a human-in-the-loop approach.

To address these challenges, we introduce **Cli**nical predictive **M**odel **B**uilder with **D**ata-**C**entric AI (**CliMB-DC**), a human-guided data-centric framework for LLM co-pilots. Building on the CliMB[1] ecosystem which focuses on democratizing model building, we advance upon it and address its limitations much like other co-pilots by integrating advanced data-centric tools, along with a novel LLM-driven reasoning process to enable robust, context-aware data processing for real-world ML challenges. Specifically, CliMB-DC introduces a novel, multi-agent reasoning system that combines a strategic coordinator for dynamic planning and adaptation with a specialized worker agent for precise execution. Domain expertise is then systematically incorporated to guide reasoning using a human-in-the-loop approach. Where CliMB established the foundation, CliMB-DC advances this vision by enabling sophisticated reasoning about data quality, integrity, and domain-specific constraints—essential capabilities for developing trustworthy ML systems when analyzing complex, real-world data.

Our contributions are as follows:

- **Taxonomy of Challenges**: We formalize a taxonomy of data-centric challenges that co-pilots need to address.

- **Data-Centric Tools**: We integrate state-of-the-art, data-centric tools into an extensible and open-source framework. The broader accessibility for non-technical domain experts to these data-centric tools allows them more options when tailoring their data management accordingly. It additionally provides an opportunity for the data-centric ML research community to incorporate new tools or validate their tools more easily.

- **Human-in-the-Loop Alignment**: We implement a human-in-the-loop system to ensure contextual alignment of data processing actions with domain-specific requirements. Moreover, we are able to incorporate domain expertise through natural language interaction, allowing experts to guide and assess data transformations without requiring coding experience.

- **Multi-Agent Planning and Reasoning** : We introduce a novel multi-agent reasoning approach that combines a strategic coordinator agent with a specialized worker agent, enabling sophisticated planning and adaptation of data-centric workflows.

- **Empirical Case Studies**: We conduct empricial case studies on real-world healthcare data, demonstrating where existing co-pilots fall short in handling the complexities of real-world data and illustrate the advantages of our approach.

CliMB-DC represents a significant step toward democratizing ML for non-technical domain experts, while ensuring the responsible and effective use of data-centric AI tools. By combining automation with expert oversight, our framework enables robust ML development that respects domain-specific knowledge. In general, the target audience for the CliMB-DC framework is broad, encompassing a wide range of users, including:

---

1. CliMB is a preliminary co-pilot version of CliMB-DC. The technical report can be found at (Saveliev et al., 2024). Extensions are introduced in Sec. 4.

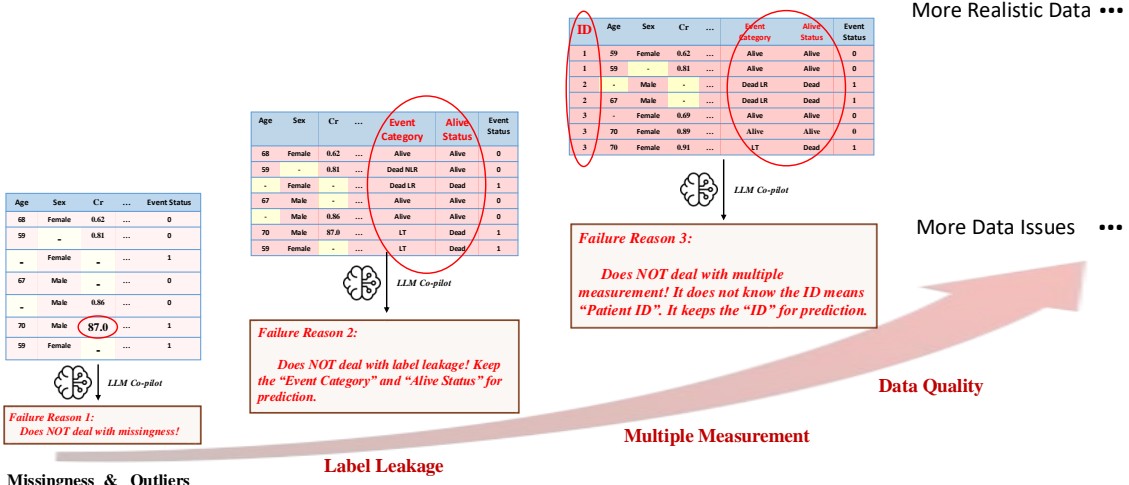

Figure 1: Illustrative examples of potential data issues in real-world healthcare scenarios, highlighting challenges at various levels and demonstrating how the current LLM co-pilot struggles to address these issues.

- **Non-Technical Domain Experts**: CliMB-DC has the potential to empower non-technical domain experts across diverse domains and application settings, including medical researchers, biostaticians, epidemiologists, social scientists, business analysts, policy makers, environmental scientists, education researchers and more. In particular, CliMB-DC enables these varied non-technical domain experts to seamlessly harness data-centric tools for research on their own datasets through an intuitive, user-friendly interface. We note that while we contextualize and instantiate CliMB-DC as a tool for healthcare, we envision that such a system could be relevant to non-technical domain experts in other data-driven domains such as finance, environmental management, education etc.

- **Data-Centric Researchers**: CliMB-DC provides a platform for data-centric researchers to effortlessly compare existing tools, integrate and validate new ones, accelerating the advancement of data-centric AI.

- **ML Researchers**: CliMB-DC can enable ML researchers across various fields to leverage state-of-the-art, data-centric tools for data preprocessing, cleaning, and assessment, simplifying and streamlining the ML research process.

## 2 Taxonomy of Data-centric Issues Facing Co-pilots

Recent advancements in LLM-based agents used in co-pilots have largely concentrated on code generation for model-centric issues, such as algorithm selection, hyperparameter tuning and performance evaluation. These processes take on datasets that have been assumedly clean (e.g., outliers removed, missingness assessed and handled, data errors removed, etc.) and problem setups that are well-defined for an ML task. However, transforming raw, sometimes disorganized, real-world datasets into clean, structured ones, while at the same

time defining a clear problem setup is not necessarily trivial and can be complex, particularly for non-technical domain experts with limited experience in data science. Such data-centric challenges are precisely the area where co-pilots are expected to provide significant support, yet have been overlooked.

**Motivated examples from healthcare.** In healthcare scenarios, it is common for some variables collected during data acquisition to be highly correlated with the outcome or to have been measured only after the outcome occurred. Including such variables in predictive models can lead to label leakage, compromising the model's validity. Consequently, these variables must be carefully excluded during model construction. As illustrated in Figure 1 (middle), current LLM co-pilot fail to exclude variables such as "Event Category" and "Alive Status", which are highly correlated with the outcome "Event Status". Including these variables results in exceptionally high predictive performance, which is a misleading conclusion for users. Similarly, healthcare datasets often contain multiple records for a single patient, as one patient could come to the hospital multiple times for follow-ups or during a chronic condition. However, current LLM co-pilots do not automatically perform aggregation to handle such cases. Being unable to appropriately account for these data structures can result in severe label leakage and render the problem setup meaningless, as demonstrated in Figure 1 (right).

Beyond label leakage, data-centric challenges in ML —including issues with data quality, preprocessing, and curation—are particularly pronounced in healthcare. These datasets are often collected by clinicians with limited data science expertise, rather than by experienced data scientists. Some datasets are retrieved from bioinformatic pipelines, which could have problems with certain reads or even produce invalid measurements. As a result, data are frequently incomplete and noisy, but usually in a context-dependent manner. The complexity when processing these data necessitates domain-specific expertise, assessment and handling. However, such challenges remain under-explored in the field of co-pilots.

**Key perspectives for ensuring reliable LLM co-pilots.** In this work, we present a formalized taxonomy of key issues that LLM co-pilots must address to enable reliable deployment in healthcare scenarios. Our taxonomy follows a bottom-up approach, drawing on a broad survey of literature where these challenges have been extensively documented and analyzed (Zadorozhny et al., 2022; Avati et al., 2021; Estiri and Murphy, 2019; Tomašev et al., 2019; Ghassemi et al., 2020; Beaulieu-Jones et al., 2017; Ferri et al., 2023; Singh et al., 2021; Haneuse et al., 2021). After synthesizing insights from these diverse studies and their practical applications, we present a structured taxonomy, highlighting the most pressing data-centric challenges affecting ML workflows. As shown in Table 1, these perspectives address both data-centric and model-centric aspects.

On the *data-centric* side, we highlight elements related to data formatting, as well as statistical (both training and test). When an LLM co-pilot fails to address these data issues effectively, it can lead to a range of problems. These include issues with the final ML model (e.g., overfitting, model bias, poor generalization, and limited interpretability) along with flaws in experimental setups (e.g., improper formulation of the problem and label leakage (see case study 1 and 2 in Section 6.2)).

While not included in the table, we also note there do remain model-centric challenges. While algorithm selection, hyperparameter tuning, and performance evaluation, have been frequently discussed and relatively well-covered in recent LLM co-pilots, there should also be a

Table 1: Taxonomy of key data-centric challenges frequently encountered in healthcare machine learning pipelines. While not exhaustive, these categories represent a significant fraction of issues that co-pilots must address to ensure strong predictive performance, robustness, fairness, and clinical feasibility.

| Category | Issues | Description | References highlighting issue | Resultant Issues |
|---|---|---|---|---|
| Data-Centric (Formatting) | Multiple measurements | Challenges from datasets including multiple observations for a single individual, requiring aggregation and standardization. | (Tschalzev et al., 2024; Liu et al., 2024; Oufattole et al., 2024; Sett et al., 2024) | Ill-posed problem setup, Temporal misalignment, Potential data inflation |
| | Multiple files | Datasets from different sources/periods of time need to be correctly aggregated or harmonized across files. | (Balagopalan et al., 2024; Lehne et al., 2019; Nan et al., 2022; Schmidt et al., 2020) | Ill-posed problem setup, Inconsistent representation, Duplication risk |
| | Inconsistent data | Data might be inconsistent based on units or how data might be represented. | (Rychert, 2023; Monjas et al., 2025; Szarfman et al., 2022) | Ill-posed problem setup, Label leakage, Reduced reproducibility |
| | Data extraction | Data might be stored in heterogenous text fields and needs to be extracted as features. | (Bao et al., 2018; Zhao, 2019; Hahn and Oleynik, 2020) | Ill-posed problem setup, Inconsistent representation |
| | Feature redundancy | Multiple features conveying similar information in a dataset. | (Chicco et al., 2022; Apicella et al., 2024; Meng et al., 2022; Sasse et al., 2023) | Poor generalization, Poor interpretability, Label leakage |
| Data-Centric (Statistical - Train) | Outliers | Extraordinary values (leading to soft outliers) or mistakes in the data creation process (possibly leading to hard outliers). | (Zadorozhny et al., 2022; Avati et al., 2021; Estiri and Murphy, 2019) | Overfitting, Misleading performance metrics, Potential data misinterpretation |
| | Label leakage | Features can include future information or tests dependent on the outcome, or datasets can have multiple correlated outcome variables. | (Tomašev et al., 2019; Ghassemi et al., 2020) | Ill-posed problem setup, Over-optimistic performance, Failed clinical deployment |
| | Missingness | Missing values caused by not being recorded (MCAR), later feature aggregation (MAR), or differing clinical practices (MNAR). | (Beaulieu-Jones et al., 2017; Ferri et al., 2023; Singh et al., 2021; Haneuse et al., 2021) | Imputation risk, Model bias, Reduced external validity |
| | Noisy labels | Incorrect labels caused by erroneous annotation, recording mistakes, or difficulty in labeling. | (Yang et al., 2023; Wei et al., 2024; Boughorbel et al., 2018) | Poor generalization, Compromised interpretability, Unstable model calibration |
| | Data valuation | General data quality issues impacting model performance. | (Bloch et al., 2021; Enshaei et al., 2022; Tang et al., 2021; Pandl et al., 2021) | Poor generalization, Suboptimal performance, High curation overhead |
| Data-Centric (Statistical - Test) | Subgroup challenges | Poor performance or generalization on certain subgroups (in-distribution heterogeneity). | (Oakden-Rayner et al., 2020; Suresh et al., 2018; Goel et al., 2020; Cabrera et al., 2019; van Breugel et al., 2024) | Poor generalization, Fairness concerns |
| | Data shift | Changes due to novel equipment, different measurement units, or clinical practice evolution over time. | (Pianykh et al., 2020; Koh et al., 2021; Patel et al., 2008; Goetz et al., 2024) | Poor generalization, Model bias, Need for continuous monitoring |

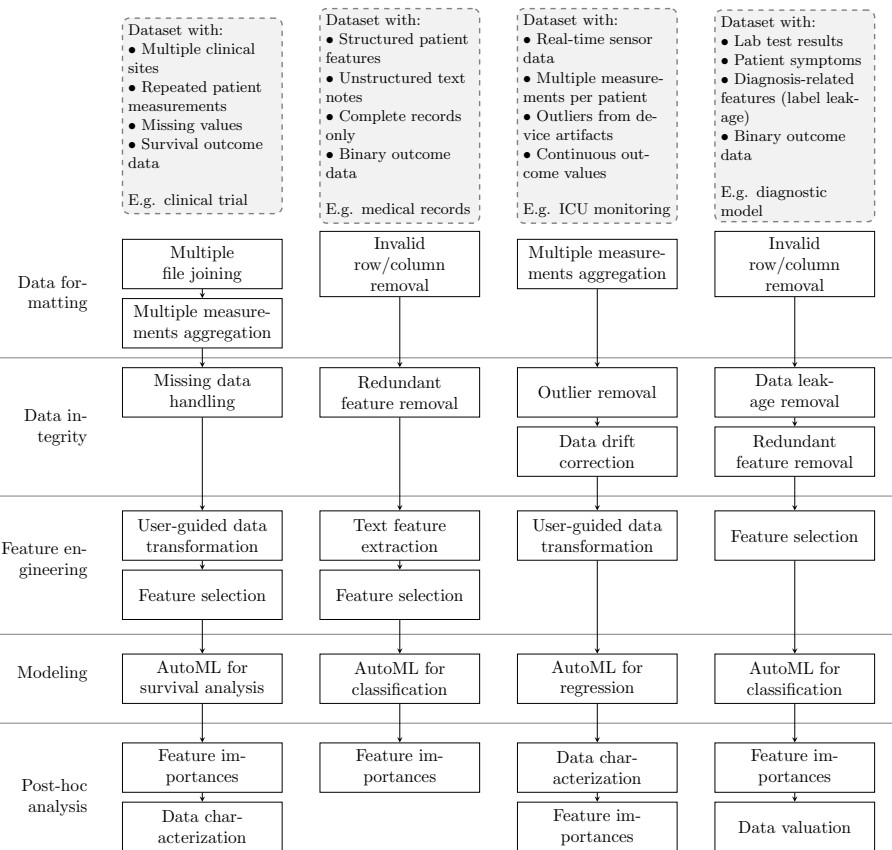

Figure 2: Addressing real data challenges is complex and requires multi-step reasoning.

focus on *domain-specific model classes* and *model interpretability*. Different from typical data science tasks that mainly focus on classification and regression: domain-specific model classes account for temporal dependencies, hierarchical structures, and clinical context, ensuring that models are both accurate and practically applicable. These issues arise frequently in data from healthcare settings. For instance, specialized models are designed for survival analysis, a critical and widely applied task in healthcare. Moreover, the role of interpretability is to ensure that predictive models can provide transparent and actionable insights, which is crucial for enabling clinicians to trust and validate their decision-making.

Our taxonomy consequently offers a systematic foundation for challenges that co-pilots should address and hence should impact the design and evaluation of LLM co-pilots. Specifically, we posit that the structured taxonomy will enable the development of co-pilots that are better equipped to handle real-world data issues, ultimately fostering more reliable, interpretable, and impactful ML systems in high-stakes domains like healthcare.

**Challenges vary by problem and context.** While the taxonomy of challenges describes each issue in an isolated manner, real-world scenarios often require a more integrated approach. When a co-pilot addresses a user's task, the challenges are inherently problem- and context-dependent, requiring end-to-end consideration. As illustrated in Figure 2, there can be multiple data issues, which must be handled in a nuanced manner, thereby making real-world applications complex. Consequently, systems must be capable of reasoning about these challenges autonomously, while gathering and considering expert human feedback.

Table 2: Comparison of different co-pilots along different dimensions.

| Perspective | Capability | DS-Agent | AutoGen | Data-Interpreter | OpenHands | CliMB-DC (Ours) |
|---|---|---|---|---|---|---|
| Components | Data-centric tools | ✗ | ✗ | ✗ | ✗ | ✓ |
| | Clinical models | ✗ | ✗ | ✗ | ✗ | ✓ |
| Expert Input | Static integration | ✓ | ✗ | ✓ | ✓ | ✓ |
| | Dynamic integration | ✗ | ✗ | ✗ | ✓ | ✓ |
| Data-centric Reasoning | Setup refinement | ✗ | ✗ | ✗ | ✗ | ✓ |
| | Performance refinement | ✗ | ✗ | ✗ | ✗ | ✓ |
| Pipeline | End-to-end automation | ✗ | ✓ | ✓ | ✓ | ✓ |
| | Replanning | ✗ | ✗ | ✗ | ✗ | ✓ |
| | Backtracking | ✗ | ✗ | ✗ | ✗ | ✓ |
| | Code refinement | ✓ | ✓ | ✓ | ✓ | ✓ |

## 3 Related Work

This work engages not only with LLM-based interpreter tools, but also with the broader area of data-centric AI research. Appendix A provides extended related work.

**LLM-Based Co-pilots.** The rapid advancements in LLMs have paved the way for various stages of ML and data science workflows to be automated by co-pilots and code interpreters that leverage the reasoning and code generation capabilities of LLMs (Tu et al., 2024; Hollmann et al., 2024; Low and Kalender, 2023). These tools allow users to specify their requirements for data science pipelines via natural language and thus offer greater flexibility compared to traditional AutoML systems. Prominent examples include systems that chain task execution (e.g., AutoGPT, DS-Agent (Guo et al., 2024)), modular frameworks for multi-step reasoning (e.g., OpenHands (Wang et al., 2024)), and graph-based workflow decomposition (e.g., Data Interpreter (Hong et al., 2024)). We provide a summary of these LLM co-pilots in Table 2, along with a detailed analysis of the co-pilots in Appendix A.1.

Despite recent progress, significant challenges remain in addressing the complex, data-centric aspects of real-world ML workflows. Many co-pilots operate within predefined pipelines or task hierarchies, making them ill-suited for dynamic, data-centric workflows. Furthermore, while these systems excel in automating code generation, they often lack mechanisms for robust data reasoning, such as diagnosing data issues, incorporating domain-specific knowledge, or addressing contextual nuances (e.g., data leakage or feature importance validation). These gaps are especially pertinent to real-world datasets that commonly exhibit variability and noise-common to data from healthcare settings (as summarized in Table 1).

Additionally, the effectiveness of these co-pilots to empower non-technical domain experts, particularly in healthcare, remains a significant challenge. Healthcare data is often characterized by heterogeneity, complexity, and susceptibility to biases and data quality issues. Hence, a co-pilot blindly applying generic data processing techniques to raw clinical data can lead to the introduction of errors and the loss of important clinical information. For example, detecting and removing outliers based on the percentile of a variable distribution might remove extreme lab values that are clinically meaningful, as they could represent a critical underlying condition rather than noise. In another example, correcting suspected label errors without domain-specific knowledge risks obscuring meaningful patterns or rare cases that are needed in downstream decision-making. These challenges underscore that when using co-pilots with non-technical domain experts, there is a need for co-pilots to reason and update via expert human guidance along with incorporating data-centric tools.

Among existing frameworks, *OpenHands* (Wang et al., 2024) and *Data Interpreter* (Hong et al., 2024) are the closest to incorporating data-centric aspects and are particularly relevant due to their emphasis on multi-step reasoning and dynamic task execution.

**Challenges with Existing Co-pilots.**  Despite the progress demonstrated by OpenHands, Data Interpreter, and similar systems, several key challenges remain (C1-C4):

- **(C1) Overlooking Data-Centric Challenges:** Existing co-pilots often overlook data quality issues such as multi-measurements, noise, outliers, and missingness. In particular, they do not integrate state-of-the-art data-centric tools. They also fail to incorporate domain-specific reasoning for tasks requiring contextual interpretation, such as deciding how to deal with multiple measurements or whether a statistical anomaly is meaningful or erroneous. The integration of human expertise is vital for this contextual reasoning.

- **(C2) Static Workflow Architectures:** Many systems operate with predefined task structures, making them ill-suited for data science workflows which depend on the unique challenges in the data or can be dynamically influenced via human expertise.

- **(C3) Healthcare-Specific Challenges:** The inability of these systems to contextualize healthcare data poses risks to using currently available frameworks. Some examples already mentioned include erroneous exclusion of clinically meaningful extreme lab values or data redundancy when retrieving data from electronic medical records or bioinformatic pipelines. Again, the integration of human expertise along with data-centric tools is vital in this regard.

- **(C4) Shallow Reasoning:** While these systems excel at automating task execution, they lack mechanisms for higher-level reasoning about data, such as validating correlations, diagnosing feature leakage, or ensuring robustness after data transformations.

**Data-centric AI.**  Data-centric AI represents a paradigm shift in ML in which assessing and improving the quality of the data are prioritized over model-specific tasks (Liu et al., 2022; Liang et al., 2022; Zha et al., 2023; Whang et al., 2023; Seedat et al., 2024). This paradigm has gained increasing importance within the ML community and has led to advances in methods and tools to systematically address issues, such as mislabeled samples (Seedat et al., 2023b; Northcutt et al., 2021a; Pleiss et al., 2020), missing data (Jarrett et al., 2022; Stekhoven and Bühlmann, 2012), outliers (Zhao et al., 2019; Yang et al., 2022), data leakage (Mitchell et al., 2019; Seedat et al., 2024), and data drifts (Cai et al., 2023; Liu et al., 2023). These approaches have demonstrated improvements in model generalization in the context of real-world scenarios characterized by noisy and heterogeneous data.

Despite the benefits of data-centric AI being demonstrated, existing LLM-based co-pilots adopt a model-centric perspective-focusing on automation of the model building pipeline, while neglecting the underlying data challenges. These limitations make existing co-pilots less effective for real-world applications where data quality directly impacts further modelling. We posit that the inclusion of data-centric AI tools in LLM-based co-pilots could significantly enhance their utility by automating the process of identifying data issues, improving dataset quality, and ensuring robust ML workflows. However, the autonomous application of these tools without contextual oversight can have unintended consequences.

Consequently, we advocate that data-centric AI tools are integrated into LLM-based co-pilots, while emphasizing the importance of the human-in-the-loop to contextualize and guide their usage.

## 4 CliMB-DC: An LLM Co-Pilot from a Data-Centric Perspective

**Problem Setting.** Denote $D = \{(\mathbf{x}^i, y^i)\}_{i=1}^n$, where $\mathbf{x} = (x_1, \ldots, x_p)$ with $x_d \in \mathcal{X}_d$ and $y \in \mathcal{Y}$, be a well-curated, "ML-ready" dataset suitable for training a given ML model (including an AutoML model) and achieving optimal target performance. Here, we consider a general scenario including both the supervised setting depending on the label types – such as classification $\mathcal{Y} = \{1, \ldots, C\}$, regression $\mathcal{Y} = \mathbb{R}$, and time-to-event analysis $\mathcal{Y} = (\{0, 1\}, \mathbb{R}_{\geqslant 0})$.

*Data corruption faced in practice.* In real-world healthcare scenarios datasets have numerous challenges as discussed. Additionally, since non-technical domain experts have limited expertise in data science, it often results in uncurated datasets, denoted as $\tilde{D} = (\tilde{\mathbf{x}}^i, \tilde{y}^i)_{i=1}^{\tilde{n}}$, where $\tilde{\mathbf{x}} = (\tilde{x}_1, \cdots, \tilde{x}_{\tilde{p}})$ with $\tilde{x}_d \in \tilde{\mathcal{X}}_d$. These datasets are subject to various data-centric issues, as highlighted in Table 1. If left unprocessed, such issues can lead to undesired failures or suboptimal performance in downstream ML models. To clarify this concept, we formalize how a well-curated dataset, $D$, can be (unknowingly and unintentionally) transformed into an uncurated dataset, $\tilde{D}$, through a series of $L$ data corruption processes during real-world data collection.

$$\tilde{D} = g(D) = g_L \circ \cdots \circ g_1(D), \tag{1}$$

where $g_\ell$ represents the corruption applied at the $\ell$-th step. A well-curated dataset, $D$, can be corrupted in numerous ways, impeding the optimal performance and clinical impact of ML models. Based on our taxonomy in Table 1, we categorize the prominent data-centric issues commonly encountered in healthcare datasets, each representing a specific type of corruption function.

*Ideal data-centric curation.* Suppose it is feasible to revert the data corruption process applied to the well-curated dataset, $D$, from the given uncurated dataset, $\tilde{D}$. Ideally, the goal is to construct a series of $L$ data curation functions, $f_1, \ldots, f_L$, where each curation function is specifically designed to revert the corresponding data corruption function applied to $D$, i.e., $f_\ell = g_\ell^{-1}$.

*Domain-specific model learning.* Once the dataset is curated, the objective is to select an appropriate, context-dependent model class and train an ML model that achieves strong generalization performance for the user-defined task descriptions.

Then we move on to introduce in detail our proposed LLM co-pilot designed through a data-centric lens, named **Cli**nical predictive **M**odel **B**uilder with **D**ata-**C**entric AI (**CliMB-DC**) [2]. See Figure 3 for the overall architecture.

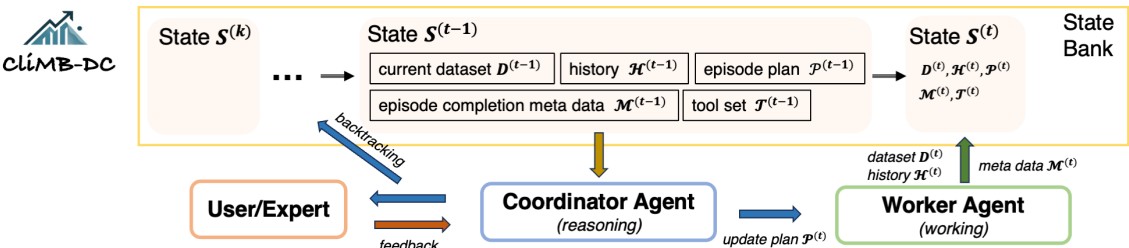

Figure 3: The overall architecture and workflow of CliMB-DC, which primarily consists of three entities that interact with the the evolving state bank: (i) a coordinator agent responsible for reasoning and planning, (ii) a worker agent for code writing and execution, and (iii) the user or human experts.

### 4.1 Overall Architecture

Recall that given an initial dataset $\mathcal{D}_0$, our goal is to find an optimal sequence of transformations $\mathbf{f} = (f_1, \ldots, f_L)$ that yields a curated dataset $\mathcal{D}^*$ suitable for downstream ML tasks. Each transformation $f_i \in \mathcal{F}$ is selected from a space of possible operations, guided by both LLM reasoning and expert feedback. The curated dataset is then used to select and train domain-specific ML models for prediction.

Our framework, CliMB-DC (see Figure 3), addresses challenges (*C1-C4* in Section 3) faced by the existing co-pilot through a dual-agent architecture that combines the strengths of LLM-based reasoning with human domain expertise.

The system consists of a high-level *coordinator agent*, responsible for managing the overall data processing strategy, and a specialized *worker agent*, tasked with executing specific data transformations. This separation of responsibilities enables CliMB-DC to maintain strategic oversight while ensuring operational efficiency. Additionally, the framework supports continuous integration of expert feedback, facilitating iterative refinement of the strategy. As shown in Figure 3, the CliMB-DC workflow involves three entities (**coordinator** and **worker** agents, and the **user/expert**) interacting with the evolving **state bank** (which includes tool sets pulled from the **tool registry**)[3]; each aspect is elaborated on below:

**State bank.** The state bank stores the system states at each time step $t$. Formally, the system state $\mathcal{S}$ at time $t$ is defined by $\mathcal{S}^{(t)} = \{D^{(t)}, \mathcal{H}^{(t)}, \mathcal{P}^{(t)}, \mathcal{M}^{(t)}, \mathcal{T}^{(t)}\}$, where $D^{(t)}$ represents the current dataset state, $\mathcal{H}^{(t)}$ captures the interaction history including expert feedback, $\mathcal{P}^{(t)}$ maintains the dynamic episode plan, $\mathcal{M}^{(t)}$ tracks episode completion metadata, and $\mathcal{T}^{(t)}$ contains the available data-centric and modeling tools that can be used by the worker agent.

**Tool registry.** Recent advances in data-centric AI have led to the development of a variety of methods and tools from across the community. CliMB-DC integrates a large variety of diverse data-centric (and model-centric tools) from across the literature (see Table 3) which are available to the worker agent to utilize. Although not exhaustive, the current set of tools covers a diverse set of scenarios linked to the data-centic challenges taxonomy in Section 2.

---

2. Code found at: https://github.com/vanderschaarlab/climb/tree/climb-dc-canonical

3. Architecturally, the *coordinator* and *worker* agents and the *tool registry* are the key components of CliMB-DC. The state bank emerges through repeated agent/user interaction with the system.

Table 3: Overview of tools available in CliMB-DC. This ensures data/model-centric tools are accessible to non-technical domain experts, while also providing data-centric ML researchers a platform for tool impact.

| Tool class | Available tools |
|---|---|
| Data understanding | Descriptive statistics, Exploratory data analysis (EDA), Feature selection (Remeseiro and Bolon-Canedo, 2019) |
| Feature extraction (from text) | spaCy Matcher |
| Data characterization | Data-IQ (Seedat et al., 2022a),TRIAGE (Seedat et al., 2023a) |
| Missing data | HyperImpute (Jarrett et al., 2022) |
| Data valuation | KNN-Shapley (Jia et al., 2019) |
| Data auditing (outliers) | Confident Learning (Cleanlab) (Northcutt et al., 2021b,a) |
| Data imbalance | SMOTE (Chawla et al., 2002) |
| Model building | AutoPrognosis 2.0 (Imrie et al., 2023) (supports regression, classification, survival analysis) |
| Post-hoc interpretability | Permutation explainer (Breiman, 2001), SHAP explainer (Lundberg and Lee, 2017), AutoPrognosis 2.0 subgroup analysis (Imrie et al., 2023) |
| Test time risk or failure analysis | Data-SUITE (Seedat et al., 2022b), SMART Testing (Rauba et al., 2024) |

Moreover, as outlined in Section 5, we illustrate the extendable nature of the framework to easily integrate new tools from the ML community.

**Coordinator agent.** The Coordinator agent is the strategic planner of the system, responsible for maintaining a high-level view of the data processing pipeline and making decisions about task sequencing (i.e. the plan). It implements a three-stage reasoning process that continuously evaluates progress, identifies potential issues, and adapts the processing strategy based on both automated metrics and expert feedback. Operating through reasoning process $\pi_{\mathcal{C}}$, the Coordinator maps the current system state to the next-step strategic decisions/processing decision (i.e. plan $\mathcal{P}^{(t+1)}$):

$$\pi_{\mathcal{C}} : \mathcal{S}^{(t)} \to \mathcal{P}^{(t+1)}. \tag{2}$$

The detailed reasoning approach is demonstrated in Section 4.2.

**User/Expert integration.** Domain expertise is systematically integrated throughout the process through feedback that ensures all transformations align with domain-specific requirements and constraints. This integration occurs through a natural language feedback mechanism that evaluates proposed transformations and enriches the system's understanding of the domain. Additionally, it captures domain knowledge that enhances the future reasoning and decision making by the coordinator agent, creating a continuous learning loop that improves the system's performance over time. The detailed interaction mechanism is introduced in Section 4.2 (see Equation 5 in *State Observation*).

**Worker agent.** The Worker agent acts as the system's execution engine, translating high-level plans from the coordinator agent into concrete data transformations instantiated in code. It combines LLM capabilities with specialized data-centric tools to implement precise, context-aware transformations while maintaining interactions to integrate information from

domain experts. The Worker's execution process is formalized as:

$$\pi_{\mathcal{W}} : (\mathcal{S}^{(t)}, \mathcal{P}^{(t)}, \mathcal{T}^{(t)}) \rightarrow (\mathcal{H}^{(t+1)}, \mathcal{D}^{(t+1)}, \mathcal{M}^{(t+1)}) \tag{3}$$

where $\mathcal{S}^{(t)}$ represents the current state, $\mathcal{T}^{(t)}$ indicates the current selected/available tool, and $\mathcal{D}^{(t+1)}$ denotes the resultant (new) dataset state, $\mathcal{H}^{(t+1)}$ and $\mathcal{M}^{(t+1)}$ denote the updated history records. The Worker operates at a granular level, focusing on individual data processing episodes and ensuring each transformation aligns with both technical requirements and domain expertise.

## 4.2 Details of CliMB-DC's Reasoning Process

Before introducing CliMB-DC's reasoning approach, we first highlight the challenges in our specific scenarios faced by an alternative approach — Monte Carlo Tree Search (MCTS) (Coulom, 2006; Rakotoarison et al., 2019).

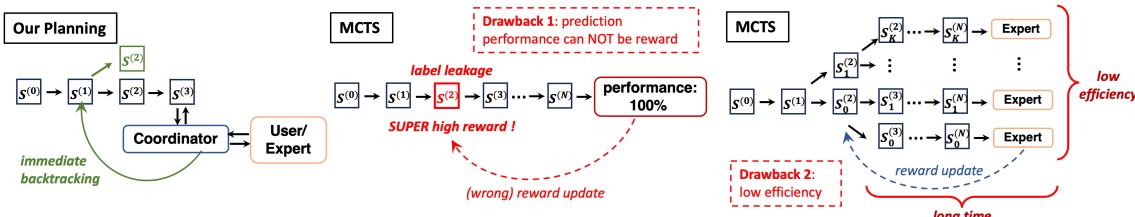

Figure 4: Challenges of Monte Carlo Tree Search (MCTS). We highlight two key drawbacks of MCTS. First, prediction performance cannot serve as a reliable reward, as it may favor data issues such as label leakage or meaningless problem setups (middle). Second, MCTS suffers from low efficiency, requiring experts to endure long waiting times and evaluate a large number of trials (right). In contrast, CliMB-DC's proposed reasoning approach enables immediate backtracking and replanning, significantly enhancing efficiency.

**Challenges of MCTS.** Monte Carlo Tree Search (MCTS) is a commonly used reasoning and planning mechanism that generates random paths to explore and evaluate potential plans based on a reward function. (Coulom, 2006; Rakotoarison et al., 2019) However, the complex nature of real-world data issues introduces several critical challenges, significantly limiting the practicality of MCTS in these contexts.

While existing co-pilots like OpenHands Wang et al. (2024) or DataInterpreter Hong et al. (2024) do not use MCTS, we provide this conceptual comparison to illustrate why MCTS, a common planning approach based on exhaustive tree exploration (as often used in reinforcement learning or agent simulations) is unsuitable in data-centric ML settings. We highlight the key challenges as follows:

- **Lack of intermediate reward model**: MCTS depends on a well-defined reward model. However, in our setting, there is no clear reward model, particularly for all the intermediate states that may arise. Even human experts are unable to provide such detailed rewards. For instance, given a dataset, it is challenging for experts to evaluate all the data issues listed in Table 1. As a result, MCTS would require complete

model training and evaluation to obtain reward signals, making iterative data curation computationally prohibitive.

- **Prediction performance is unsuitable as the final reward**: Another significant challenge for MCTS in our scenarios is that prediction performance cannot be directly used as the final reward. It is critical to first ensure that the entire data processing pipeline is valid and free from issues such as label leakage, which could render performance metrics unreliable. For example, as shown in Figure 4 (middle), if the state $\mathcal{S}^{(2)}$ introduces label leakage, relying solely on performance to determine the reward would assign an artificially high reward to this node. However, such a state should be avoided in the final path.

- **Low efficiency**: A further challenge is the low efficiency of the process. Since prediction performance is inadequate as a final reward, expert evaluation may be necessary. However, MCTS involves random exploration and requires numerous steps to transform a raw dataset into a well-trained prediction model. As shown in Figure 4 (right), this results in lengthy trials, and the large number of trials exacerbates the inefficiency. Additionally, in many scenarios, users may be unable to examine the details of all trials due to time constraints or limited expertise in data science and clinical domain knowledge. These limitations significantly restrict the applicability of MCTS in our scenarios.

The challenges associated with MCTS largely arise from its approach of treating all data processing steps as unknown and unexplored, attempting to navigate the entire sequence of actions, as is common in gaming scenarios. In contrast, data processing typically follows a well-established order of operations, making such exhaustive "searching" unnecessary. For example, addressing data missingness generally precedes other transformations or feature engineering steps, and exploring these well-known rules through extensive random trials is both inefficient and redundant.

A more practical alternative to MCTS is to focus on refining "local processing" within the general sequence of operations. Our framework incorporates automated planning combined with expert validation to ensure both technical quality and domain-specific appropriateness. The key innovation lies in enabling the method to backtrack after errors and consult experts when necessary, such as for decisions involving the meanings of features, handling label leakage, or determining whether to drop specific features, etc. As shown in Figure 4 (left), when combined with immediate expert feedback, the coordinator enables prompt backtracking, significantly improving efficiency.

**CliMB-DC's proposed multi-stage reasoning.** The reasoning mechanism of CliMB-DC is demonstrated in Figure 5. At time $t$, the coordinator agent takes the current state bank $\{\mathcal{S}^{(0)}, \ldots, S^{(t-1)}\}$ as input, and outputs the plan via:

$$\underbrace{P(\mathcal{P}^{(t)}|\{\mathcal{S}^{(0)}, \ldots, \mathcal{S}^{(t-1)}\})}_{\text{coordinator reasoning}} = \sum_{\mathcal{O}^{(t)}} \underbrace{P(\mathcal{O}^{(t)}|\mathcal{S}^{(t-1)})}_{\text{state observation}} \cdot \left( \sum_{\beta^{(t)}} \underbrace{P(\beta^{(t)}|\mathcal{O}^{(t)})}_{\text{backtracking}} \underbrace{P(\mathcal{P}^{(t)}|\beta^{(t)}, \mathcal{O}^{(t)})}_{\text{lookahead planning}} \right), \quad (4)$$

where

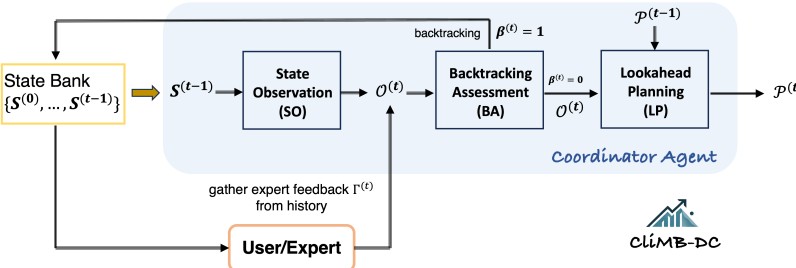

Figure 5: The framework of the coordinator agent in CliMB-DC, encompassing three parts named State Observation (SO), Backtracking Assessment (BA), and Lookahead Planning (LP).

1. **State Observation (SO)**: $P(\mathcal{O}^{(t)}|\mathcal{S}^{(t-1)})$ denotes the state observation stage, where the coordinator analyzes the project state, focusing primarily on the last state $\mathcal{S}^{(t-1)}$. And it will gather expert human feedback $\Gamma^{(t)}$ from history interactions. Therefore, $P(\mathcal{O}^{(t)}|\mathcal{S}^{(t)})$ can be formulated as:

$$P(\mathcal{O}^{(t)}|\mathcal{S}^{(t-1)}) = \sum_{\Gamma^{(t)}} P(\Gamma^{(t)}|\mathcal{S}^{(t-1)})P(\mathcal{O}^{(t)}|\Gamma^{(t)}, \mathcal{S}^{(t-1)}), \tag{5}$$

where $\Gamma_t$ denotes the expert feedback at time $t$ and the summation is a marginalization over all possible feedback signals. To inform its decision, the coordinator extracts information about the project state such as evaluating experiment outcomes, data quality metrics and expert feedback received, together denoted as $\mathcal{O}^{(t)}$. In practice, we observe a single realized feedback message from the user at each step; the marginalized form is included to make the underlying probabilistic dependencies explicit.

2. **Backtracking Assessment (BA)**: Based on the analysis, the coordinator determines if previous decisions need revision or updating. If the project is not progressing satisfactorily (e.g., due to data quality issues or expert feedback indicating problems), the coordinator identifies a backtracking point $k < t - 1$ and restores the project state to $\mathcal{S}^{(k)}$ (i.e. backtrack step), which is denoted by $P(\beta^{(t)}|\mathcal{O}^{(t)})$.

3. **$M$-Step Lookahead Planning (LP):** The coordinator evaluates the current plan, focusing specifically on the next $M$ episodes $e_t, \ldots, e_{t+M-1}$ within the plan $\mathcal{P}^{(t)}$. For each episode $e_i$, the coordinator assesses two key aspects: (i) Necessity and (ii) Appropriateness. Leveraging the history and user interactions (expert feedback) stored in $\mathcal{O}^{(t)}$, the coordinator refines the plan by excluding episodes considered unnecessary or inappropriate and incorporating new ones as needed. This process ensures the updated plan $\mathcal{P}^{(t)}$ remains aligned with the user's objectives. Specifically, this can involve the following types of updates:

   - Reordering episodes to better handle dependencies;
   - Removing unnecessary episodes;
   - Adding new episodes to address identified gaps;
   - Modifying episode parameters based on expert input.

The coordinator then issues this updated plan to the worker agent to execute at the next iteration. When the Worker agent completes an episode, control returns to the Coordinator for the next iteration of plan analysis and refinement.

**Example 1 (Demonstration of the reasoning process.)** *Let us consider the case of a dataset with missingness and how the reasoning process works. As shown in Figure 6, after loading the dataset, the co-pilot detects the missingness issue and initially plans to address it using the DropNA function, which removes all rows with missing values, resulting in the state $\mathcal{S}^{(2)}$. However, the State Observation (SO) module identifies a new problem: the reduced dataset size falls below 50, which is insufficient for subsequent processing and difficult to remedy. In response, the Backtracking Assessment (BA) module is triggered, rolling the state back to $\mathcal{S}^{(1)}$.*

*In the next step, the SO module detects the missingness issue, and the BA module is not triggered. Drawing on the history record, which indicates that using "DropNA" previously led to backtracking, the Lookahead Planning (LP) module revises the plan and selects an alternative approach—imputation—to address the missingness issue.*

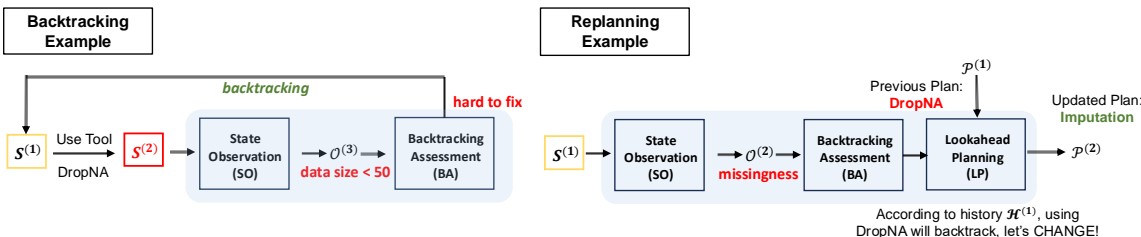

Figure 6: Example of backtracking and replanning in handling missing data, showcasing how the Backtracking Assessment (BA) and Lookahead Planning (LP) modules in the proposed reasoning approach collaborate to efficiently resolve data issues.

**Reasoning process algorithm.** We formalize this iterative process in Algorithm 1, which details the interaction between components and the progression of transformations.

### 4.3 Worker Agent

The worker agent takes the updated plan and integrates it with the available tool set (see Table 3). It then generates and executes the necessary code to complete the plan. If execution fails, the agent autonomously updates the code to ensure successful execution. Additionally, the worker agent can verify the availability of required Python packages, installing them if necessary before proceeding with execution. For the case studies in Section 6, we perform an ablation study by removing the coordinator & our tool set from CliMB-DC to emphasize the reliability of our worker agent in code generation and execution.

***Remark on worker and coordinator agents***: We clarify that the multi-agent architecture of coordinator and worker are instantiated with the same backbone LLM, however, they are differentiated via distinct prompting, role specialization, tool access, memory modules, distinct reasoning responsibilities and system state partitioning.

---
**Algorithm 1** CliMB-DC Optimization with Expert Integration

---
**Require:** Initial dataset $\mathcal{D}^{(0)}$, tools $\mathcal{T}$
**Ensure:** Curated dataset $\mathcal{D}^*$
 1: Initialize $\mathcal{S}^{(0)} = \{\mathcal{D}^{(0)}, \varnothing, \mathcal{P}^{(0)}, \varnothing, \mathcal{T}\}$
 2: $\mathcal{D}^* \leftarrow \mathcal{D}^{(0)}$
 3: **while** not converged **do**                                   ▷ Coordinator reasoning phase
 4:       $\mathcal{O}^{(t)} \leftarrow \text{STATEOBSERVE}(\mathcal{S}^{(t)})$                             ▷ see Section 4.2
 5:       $\beta^{(t)} \leftarrow \text{ASSESSBACKTRACK}(\mathcal{O}^{(t)})$
 6:       **if** $\beta^{(t)} = 1$ **then**
 7:           $(\mathcal{D}^*, \mathcal{S}^{(t)}) \leftarrow \text{RESTORECHECKPOINT}(\mathcal{H}^{(t)})$
 8:           continue
 9:       **end if**
10:       $\mathcal{P}^{(t)} \leftarrow \text{PLANNING}(\mathcal{O}^{(t)})$
11:       **while** not episode_complete **do**                        ▷ Worker execution phase
12:           $f_t \leftarrow \mathcal{W}.\text{PROPOSETRANSFORM}(\mathcal{S}^{(t)}, \mathcal{P}^{(t)}, \mathcal{T}^{(t)})$
13:           $\mathcal{D}^* \leftarrow f_t(\mathcal{D}^*)$
14:       **end while**
15:       $\mathcal{S}^{(t+1)} \leftarrow \text{UPDATESTATE}(\mathcal{S}^{(t)}, \mathcal{D}^*, \mathcal{H}^{(t)})$          ▷ Update system state
16: **end while**

---

## 5 CliMB-DC: Open-source software toolkit

**Code:** https://github.com/vanderschaarlab/climb/tree/climb-dc-canonical

Beyond usage by diverse users and improved performance, an important aspect for CliMB-DC for impact in healthcare is its role as a software toolkit to empower domain experts. Consequently, a key aspect is the open-source nature of the framework, which enables the community to contribute and integrate new tools to extend its capabilities.

To achieve this goal of empowerment for diverse users, three software challenges are vital to address: *extensibility to new tools*, *human integration*, and *support for diverse predictive tasks* in medicine, specifically *classification*, *survival analysis*, and *regression*. This enables a more robust and user-friendly system for clinical predictive modeling.

### 5.1 Extensibility to New Tools

The diversity and rapid development of data-centric tools means the framework must be capable of integrating new tooling from the community with minimal effort.

**Data-Centric Tool Support:** CliMB-DC emphasizes a data-centric AI approach by integrating specialized tools (see Table 3) that enhance dataset quality including:
- *Imputation Tools*: Frameworks like HyperImpute handle missing data with advanced iterative imputation techniques.
- *Exploratory Analysis and data quality evaluation*: Tools such as Data-IQ enable detailed subgroup analysis, data heterogeneity and noisy labels.
- *Interpretability*: Built-in post-hoc interpretability methods like SHAP and permutation explainers ensure models remain transparent and actionable.

**Tool use:** The worker uses tools through code generation, which calls Python APIs or data-centric tools using their desired software interfaces (e.g., Data-IQ, hyperimpute, shap, etc.). Tools are registered with metadata, including API signatures and expected inputs, offering controlled tool usage. In the case of custom operations, the worker writes executable Python code, which is logged and shown to the user — to allow for traceability.

**Extensibility:** (i) We have a *tool registry* which catalogs the available tools, their supported predictive tasks, and their data requirements, enabling users to easily incorporate new methods without modifying core system logic. (ii) Modular APIs allow developers to register new tools, ensuring that CliMB-DC evolves alongside data-centric advances. (iii) The Open-source architecture encourages community contributions to expand the ecosystem of available data-centric tools to the co-pilot. This enables broader accessibility to data-centric tools for non-technical domain experts thereby empowering them. Additionally, it provides an opportunity for the data-centric ML research community to easily incorporate new tools and/or validate their tools, facilitating research impact via usage on diverse applications.

## 5.2 Human Integration Through UI and Feedback

More complex ML frameworks generally require a wider range of skill sets that are often lacking by non-technical domain experts, whereas the setup of more complex biological research questions risk being misunderstood by technical domain experts. One way of minimizing the impact of these limitations is creating a user interface (UI) that allows both mutual understanding of the tasks between users and integrates specific feedback from the type of user.

**User Interface** The UI for CliMB-DC combines output from natural, conversational language, along with updates on the progress of the task pipeline accompanied with visualizations (see Figure 7). This type of interface provides non-technical domain experts the opportunity to perform tasks that they might not be able to do directly with an ML tool and technical domain experts the opportunity to examine more closely which ML procedures were effectuated.

Oftentimes, users with technical and non-technical domain expertise employ different terminology for the same task or problem at hand, obfuscating the processes needed to complete the task or solve the given problem. For example, users from the ML community might adapt a feature selection process prior to or during predictive modeling, while those from the epidemiology community would refer to this process as model building (see Table 8 for more examples). The fact that users can communicate desired processes in their natural language makes it possible to carry out the intended task, supporting its nuances and bringing about a more fluid user experience.

**Dynamic Plan Refinement via expert guidance** CliMB-DC importantly incorporates *human feedback* into its reasoning. The user can refine the data science pipeline in an iterative manner by weighing in their expertise on a variety of processes (e.g., on data transformations, feature selection, or model evaluation). The iterative feedback dynamically adjusts the plan, ensuring alignment with domain-specific goals and that outputs are clinically relevant.

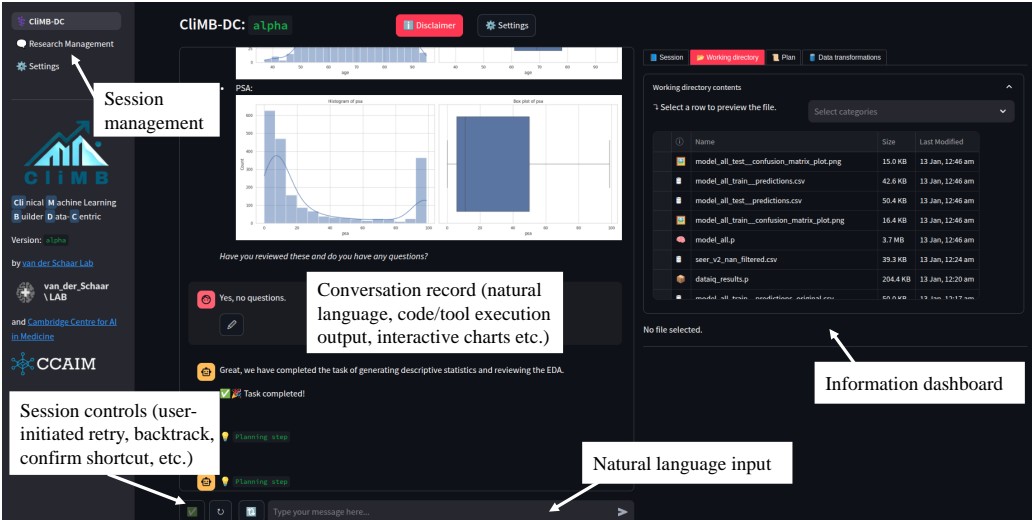

Figure 7: The user interface of CliMB-DC, which supports natural language input, multi-modal conversation record and dashboard, session controls (including user-initiated retry and backtracking), and session management across multiple conversations.

## 5.3 Support for Diverse Predictive Tasks

Clinical datasets require predictive modeling across varied tasks, including *classification*, *survival analysis*, and *regression*, each with distinct data processing and modeling needs. Recall that in carrying out these analytics, non-technical domain experts like clinical researchers, biostatisticians, epidemiologists etc, do not need to do ML analytics. Rather CliMB-DC facilitates this with domain expertise used to guide and validate the process.

## 6 Case Studies

We now empirically investigate multiple aspects of CliMB-DC to handle real-world healthcare data challenges.

1. **Does it work?**: We highlight for multiple data-centric challenges from our taxonomy, where vanilla co-pilots fail and the data-centric lens with human feedback helps.

2. **Why does it work?**: We provide an in-depth analysis via various case studies to better understand why CliMB-DC succeeds and other co-pilots fail.

**Datasets.** We employ real-world tabular healthcare datasets with varying characteristics. Specifically, different sample sizes, dimensionality, task types (classification, survival analysis) and task difficulty. These datasets reflect the following data challenges (as defined in our taxonomy): (i) Lung cancer: Data leakage (ii) PBC Dataset: Unaggregated data (based on identifiers) and (iii) Prostate cancer prediction: Ambiguous and Hard examples (mislabeled and outliers). The datasets are detailed in Appendix B.

**Baselines.** We compare CliMB-DC as discussed in the related work to Data Interpreter (Hong et al., 2024), OpenHands (Wang et al., 2024) and LAMBDA (Sun et al., 2025) as representative co-pilots. Additionally, we perform an ablation of CliMB-DC. We remove the coordinator, highlighting its value, while assessing the worker agent's reliability in code

generation and execution. This ablation is instantiated both with and without tools. We refer to these configurations as CliMB-DC (No coordinator, With tools) and CliMB-DC (No coordinator, No tools), respectively. These can be thought of as ablations which are reflective of the original CliMB. Unless otherwise stated all results are averaged over 5 runs. In addition to the results in the main paper, Appendix C and D provides a detailed analysis of the logs of interactions for CliMB-DC and the baselines.

## 6.1 Does it work?

To demonstrate the effectiveness of CliMB-DC, we conduct the following case studies on healthcare datasets using different data challenges. Below, we summarize the results and the context and importance of these datasets:

- **Case study 1:** In the dataset on predictors of lung cancer, the primary challenge involves addressing data leakage. This scenario is particularly complex as it combines survival analysis with the need to identify and handle potentially leaked information from outcome-related variables. Table 5 demonstrates based on C-index that both Data-Interpreter and OpenHands were unable to provide valid results in almost all scenarios and produced several different reasons for failures, most of which was related to data leakage. CliMB-DC was able to produce valid results without run failures.

- **Case study 2:** In the dataset on predictors of PBC, unaggregated data and potential data leakage are presented as simultaneous challenges. Table 4 demonstrates based on C-index that both Data-Interpreter and OpenHands were unable to provide vaild results, mainly due to failure to aggregate data per patient and identify data leakage. This scenario is especially relevant to healthcare settings where multiple measurements per patient are common. CliMB-DC was able to handle these issues and produce valid results, while maintaining temporal consistency and avoiding information leakage.

- **Case study 3:** In the datasets comparing predictors of prostate cancer mortality from the SEER (USA) and CUTRACT (UK) datasets, the challenge lies in handling data quality issues and data drifts, across different healthcare systems. Table 6 demonstrates based on AU-ROC that all three tools were able to produce valid results, while the accuracy and AU-ROC was slightly higher when using CliMB-DC. The results demonstrate our framework's robustness in managing dataset shifts while maintaining consistent performance across different healthcare contexts.

Table 4: Results on the PBC dataset, where the primary data challenges are addressing *unaggregated data* and *data leakage*. The prediction task in this case is *survival analysis*, a specialized and less common task compared to those typically encountered in general machine learning fields. The whole processing procedure of the proposed CliMB-DC is shown in Figure 8.

| Method | Human Assistance | Results Valid | C-Index | Failure Modes | % runs tested |
|---|---|---|---|---|---|
| Data-Interpreter | - | × | 0.789 | Failed to aggregate data per patient
Failed to identify data leakage
Failed to produce results
Failed to set up survival problem | 100%
100%
40%
20% |
| OpenHands | - | × | 0.468 | Failed to aggregate data per patient
Failed to identify data leakage
Failed to set up survival problem
Convergence issues causing task failure | 100%
100%
60%
20% |
| LAMBDA | - | × | N/A | Failed to process or load data | 100% |
| **CliMB-DC** (No Coordinator & No Tools) | - | × | 0.663 | Failed to aggregate data per patient
Failed to identify data leakage
Failed to produce results
Fail to solve convergence error | 100%
80%
60%
20% |
| **CliMB-DC** (No Coordinator & With Tools) | - | × | 0.914 | Failed to aggregate data per patient
Failed to identify data leakage | 100%
100% |
| **CliMB-DC** | - | ✓ | **0.953** | (Successful) | 100% |

Table 5: Results on the Lung Cancer dataset, where the primary data challenge is addressing *data leakage*. The prediction task in this case is *survival analysis*, a specialized and less common task compared to those typically encountered in general machine learning fields. The whole processing procedure of the proposed CliMB-DC is shown in Figure 9.

| Method | Human Assistance | Results Valid | C-Index | Failure Modes | % runs tested |
|---|---|---|---|---|---|
| Data-Interpreter | - | × | 0.625 | Failed to identify data leakage
Incorrect metric used
Failed to set up survival problem
PCA use degraded performance | 100%
20%
20%
60% |
| | Leakage features excluded | ✓
× | 0.738
0.995 | (Successful)
Label leakage reintroduced | 80%
20% |
| OpenHands | - | × | N/A | Failed to identify data leakage
Incorrect metric reported | 100%
100% |
| | Specify Cox model | × | 0.496 | Failed to identify data leakage | 100% |
| | Leakage features excluded | ✓
× | 0.500
N/A | (Successful)
Stuck in a loop | 80%
20% |
| LAMBDA | - | ✓
× | 0.689
0.887 | (Successful)
Failed to identify data leakage | 40%
60% |
| **CliMB-DC** (No Coordinator & No Tools) | - | × | 0.765 | Failed to identify data leakage
Failed to solve convergence error | 100%
40% |
| | Leakage features excluded | ✓
× | 0.809
N/A | (Successful)
Failed to test on the test file | 80%
20% |
| **CliMB-DC** (No Coordinator & With Tools) | - | × | 0.871 | Failed to identify data leakage | 100% |
| **CliMB-DC** | - | ✓ | **0.848** | (Successful) | 100% |

Table 6: Results on cross cancer mortality prediction (SEER from the USA to CUTRACT from the UK), where the primary data challenges are addressing *data quality/hardness* and *data drifts*. The prediction task in this case is *classification*, a common task in general machine learning fields. The whole processing procedure of the proposed CliMB-DC is shown in Figure 10.

| Method | Human Assistance | Results Valid | Accuracy | AUC-ROC | Failure Modes | % runs tested |
|---|---|---|---|---|---|---|
| Data-Interpreter | - | ✓ | 66.5 | 0.727 | (Successful) | 80% |
| | | ✗ | N/A | N/A | Failed in preprocessing | 20% |
| OpenHands | - | ✓ | 67.5 | 0.729 | (Successful) | 100% |
| LAMBDA | - | ✓ | 67.1 | 0.726 | (Successful) | 100% |
| **CliMB-DC** (No Coordinator& Tool Set) | - | ✓ | 67.8 | 0.749 | (Successful) | 80% |
| | | ✗ | 68.3 | 0.683 | Failed to compute AUC-ROC | 20% |
| **CliMB-DC** (No Coordinator & With Tools) | - | ✓ | 69.4 | 0.765 | (Successful) | 100% |
| **CliMB-DC** | - | ✓ | **69.9** | **0.771** | (Successful) | 100% |

## 6.2 Why does it work?

To provide a deeper understanding of how CliMB-DC can excel in specific, data-centric challenges, we describe various facets of using CliMB-DC in comparison to other co-pilots. These case studies illustrate how our multi-agent architecture, reasoning processes and human-in-the-loop feedback can provide specific advantages, and where problems when using other co-pilots arise. Figures 8 and 9 illustrate the specific reasoning and planning mechanisms through which CliMB-DC reasons, adapts the plan, engages with the domain expert and then resolves these challenges.

**Case study 1:** In many datasets from healthcare settings, multiple measurements are recorded for each patient over time, leading to unaggregated data. Baseline co-pilots treat each row as an independent patient observation, creating two issues: (1) data leakage between training and test sets when measurements from the same patient appear in both, and (2) an ill-posed modeling setup that violates the independence assumptions of survival analysis.

In contrast, as shown in Figure 8, CliMB-DC identifies the structure of the dataset, leveraging its state observation and human-guided feedback mechanisms to aggregate measurements correctly. This ensures that the modeling process aligns with the true data generation process, avoiding leakage.

**Case study 2:** Survival analysis tasks are particularly vulnerable to label leakage from other features or covariates that can compromise model validity. Specifically, time-dependent variables like "time_to_lung_cancer" inherently leak information about the outcome. Other co-pilots fail to account for such features, resulting in inflated performance metrics and compromising the model's real-world applicability and generalization.

In contrast, as shown in Figure 9, CliMB-DC's dynamic reasoning and iterative planning allows detection and mitigation of label leakage. Through domain expert feedback, the system removes problematic features like *"time_ to_ lung_ cancer"*. This ensures that the resulting models are valid and generalizable.

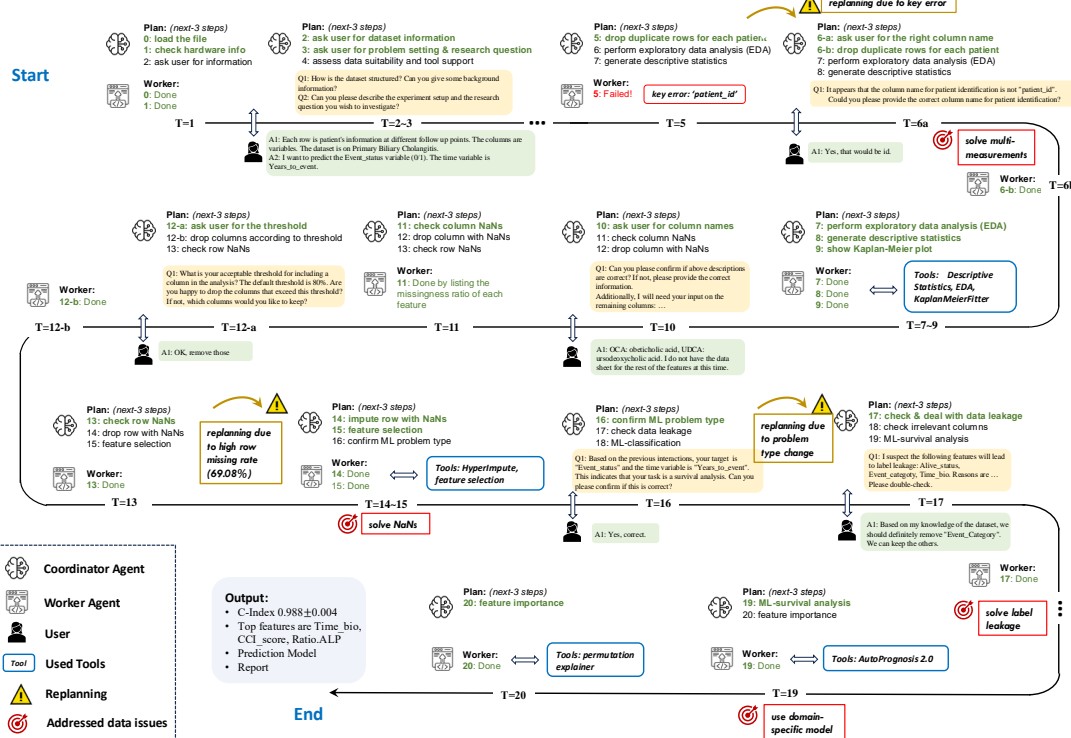

Figure 8: End-to-end session trace of CliMB-DC on the PBC dataset. Shows how the coordinator, worker, and user interact to detect and resolve real-world issues like missingness, multi-measurements, and label leakage, culminating in a survival analysis model. The agent successfully replans upon encountering and identifying: (i) multiple measurements, (ii) NaNs, (iii) label leakage and (iv) the need for a domain-specific model.

**Case study 3:** Data from healthcare settings can often have data quality challenges such as hard examples (mislabeled, heterogenous outcomes etc). These observations can affect model training (Seedat et al., 2022a) and can be sculpting or filtered from the dataset to improve generalization. In addition, when models are used across countries, as is the case in the two prostate cancer datasets from SEER and CUTRACT, distribution shift could occur.

As shown in Figure 10, CliMB-DC's dynamic reasoning and data-centric tool usage allow it to understand that data quality is a challenge, run a method for data characterization (e.g. Data-IQ (Seedat et al., 2022a)) and based on the output, engage with the human expert to remove ambiguous observations that may cause downstream problems during modeling. We show that this improves model generalization cross-domain (i.e. in different countries).

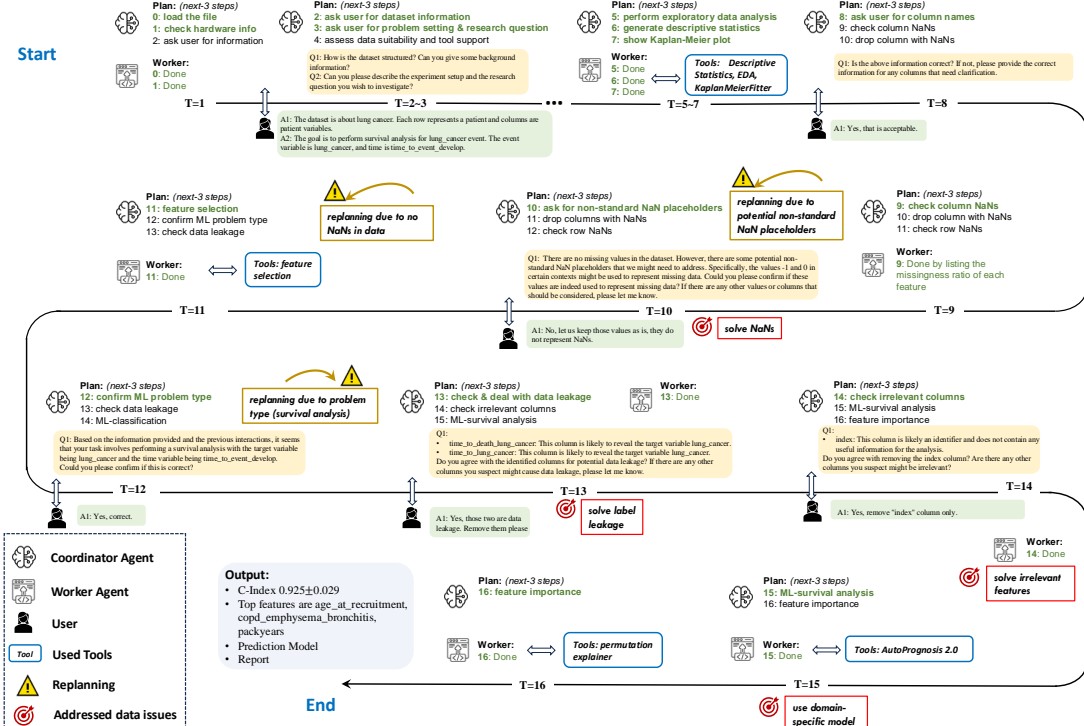

Figure 9: End-to-end session trace of CliMB-DC on the lung cancer dataset. Demonstrates the detection of label leakage from time-related features, resolution via expert feedback, the detection and resolution of irrelevant features and final model training using an appropriate survival analysis model. The example highlights the importance of domain-guided/expert-guided data curation.

## 7 Conclusion

In this work, we introduced CliMB-DC, a human-guided, data-centric framework for LLM-based co-pilots. Importantly, this framework addresses a critical gap in current LLM co-pilots: their inability to effectively handle real-world data challenges while incorporating domain expertise. Our contributions span multiple dimensions, from a taxonomy of data-centric challenges to developing a novel multi-agent architecture enabling sophisticated reasoning about data quality and processing workflows.

Our empirical evaluations highlight several key advantages of CliMB-DC when handling key data challenges, allowing it to achieve robust ML outcomes where existing co-pilots may come across problems. Beyond these technical contributions, the open-source nature of CliMB-DC encourages the broader research community to extend its capabilities, ensuring its relevance across diverse data structures and modeling applications.

By highlighting the importance of data-centric aspects to AI co-pilots, CliMB-DC represents a critical step towards democratizing ML for non-technical domain experts (in a variety of fields), ensuring that data quality and contextual understanding are central to ML workflows. We envision this framework as a foundational tool for democratizing the adoption of ML across a variety of problem settings and domains.

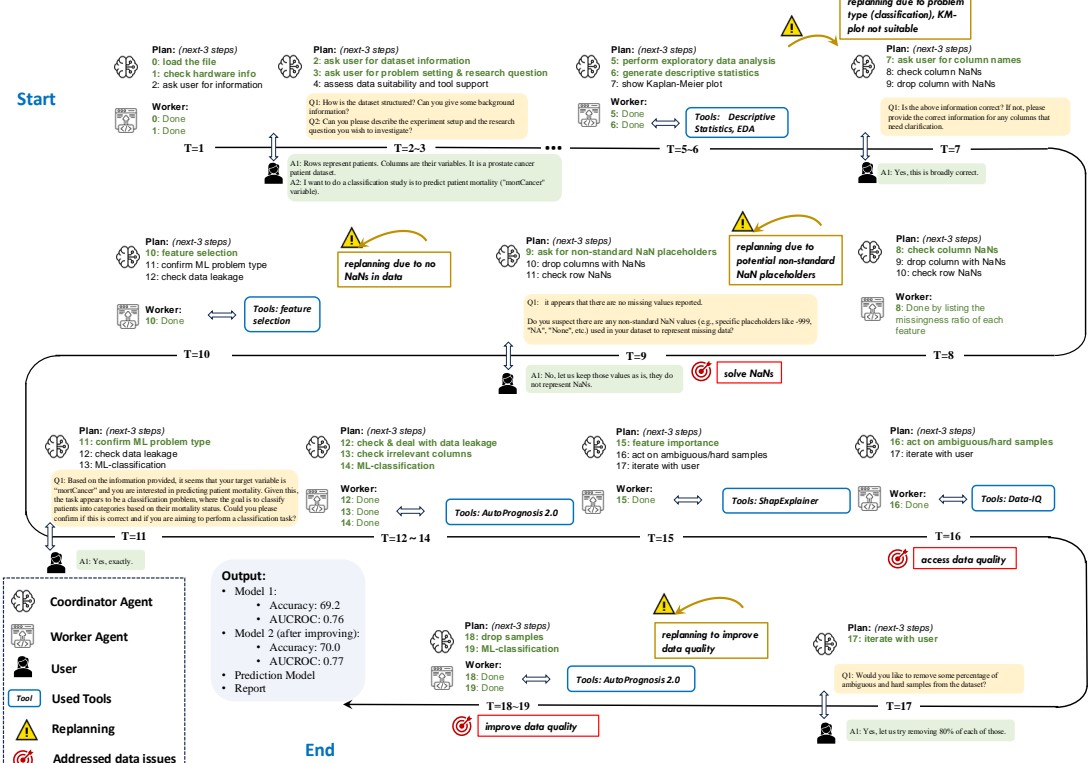

Figure 10: End-to-end session trace of CliMB-DC on cross-cohort cancer mortality task (SEER to CUTRACT). Illustrates how CliMB-DC maintains robust performance across cohorts by handling data drift and quality issues (i.e. via reasoning and tool invocation) leading to better downstream generalization on the curated datasets.

## Broader Impact Statement

CliMB-DC has significant potential implications, both positive and negative. On the beneficial side, by democratizing machine learning for non-technical domain experts (such as clinical researchers, biostatisticians, epidemiologists, social scientists, business analysts, environmental scientists, education researchers and more etc), it could accelerate the development and deployment of ML-based predictive and decision support tools. Our framework's emphasis on data-centric challenges (which is both impactful and consumes significant time), coupled with the integration of domain expertise helps ensure that resulting ML models are more reliable and relevant within a problem domain, which is crucial for safety and trust in AI.

From an ML perspective the impact is: (1) giving a platform to the data-centric ML research community to integrate their tools into our open source framework and (2) our novel multi-agent reasoning process to adapt co-pilots more dynamically and account for human feedback.

However, this democratization of ML tools also carries risks. Even with built-in safeguards, there is potential for misuse if users do not fully understand the limitations of the models or overlook important domain-specific nuances of their data. The framework could inadvertently

amplify existing biases in data if users do not carefully consider data issues. Additionally, while the human-in-the-loop approach helps mitigate risks, it relies on user expertise.

Related to the risks are also issues around privacy preservation. CliMB-DC addresses these as follows: (1) Local storage of data: All datasets used by CliMB-DC remain local to the user's machine. No dataset (raw or transformed) is uploaded to any external server. (2) Local code execution: All code execution, whether through generated Python or integrated tools, occurs locally. Hence, the actual data does not leave the user's machine. Rather, just the project state is the only thing sent to the LLM. This ensures that CliMB-DC remains compliant with common data protection regulations such as HIPAA in the United States and GDPR in the European Union, assuming appropriate local data governance.

That said, in terms of broad applicability, the CliMB-DC framework while instantiated for healthcare tasks, is applicable to non-technical domain experts in other domains such as finance, social sciences, education etc.

Finally, to maximize positive impact while minimizing risks it is important that users understand their roles and what the framework can and cannot do when using the co-pilot.

## Acknowledgments and Disclosure of Funding

We thank Robert Davis for his help with the software and Changhee Lee, Thomas Pouplin and Mikkel Werling for their helpful discussions. This work was supported by Azure sponsorship credits granted by Microsoft's AI for Good Research Lab, with special thanks to Dr. Bill Weeks, Director of AI for Health Research at Microsoft, for his support. This work was additionally supported by Microsoft's Accelerate Foundation Models Academic Research initiative. NS is funded by the Cystic Fibrosis Trust. ES is funded by Dinwoodie Charitable Company and University Hospital Southampton NHS Foundation Trust.

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

# Appendix: Towards Human-Guided, Data-Centric LLM Co-Pilots

## Table of Contents

Code: https://github.com/vanderschaarlab/climb/tree/climb-dc-canonical

## Appendix A. Extended related work

Below we provide a further assessment of LLM-based code interpreters.

### A.1 LLM-Based Code Interpreters

- **GPT-Code Interpreter** [4]**:** Aimed at simplifying tasks such as data visualization, basic modeling, and statistical analysis, this tool allows users to query and interact with datasets dynamically. However, its design is limited to single-step tasks and lacks support for multi-stage workflows, iterative refinement, or complex reasoning across interdependent tasks. Its applicability to real-world datasets with evolving requirements is minimal due to its static nature.

- **AutoGPT** [5]**:** AutoGPT generalizes task execution by chaining multiple steps through autonomous prompts. It explores iterative workflows but relies heavily on predefined task templates. This rigidity makes it ill-suited for dynamic, data-centric environments where task dependencies evolve unpredictably. Moreover, AutoGPT lacks mechanisms to diagnose or correct data quality issues during execution.

- **BLADE (Gu et al., 2024):** Designed primarily as a benchmarking framework, BLADE evaluates LLM agents on open-ended scientific analyses and decision-making tasks. It provides insights into flexibility and task accuracy but does not address robustness or the ability to adapt workflows based on intermediate results. Furthermore, its scope is confined to task execution, neglecting data-centric challenges such as feature leakage or outlier handling.

- **DS-Agent (Guo et al., 2024):** Integrating case-based reasoning (CBR) with LLMs, DS-Agent automates ML workflows by leveraging prior knowledge from human-curated cases (e.g., Kaggle). It iteratively refines workflows by incorporating execution feedback. However, its dependency on retrieved cases limits its adaptability to novel or unstructured problems. DS-Agent's reliance on historical cases also makes it less effective for workflows requiring real-time adaptability or dynamic reasoning about data.

- **OpenHands (Wang et al., 2024):** OpenHands introduces a modular architecture for multi-agent collaboration and secure task execution in sandboxed environments. Its strengths lie in its flexibility and support for multi-step workflows, including software engineering tasks and web interaction. However, it lacks built-in tools for diagnosing and resolving data-centric issues, such as missing data or noise, and offers limited support for domain-specific reasoning, which is critical for high-stakes domains like healthcare.

- **Data Interpreter (Hong et al., 2024):** Data Interpreter employs hierarchical graph-based reasoning to model workflows as interdependent tasks, allowing for iterative refinement and robust task decomposition. This makes it highly effective for structured ML pipelines. However, its reliance on predefined task graphs limits its generalization

---

4. https://platform.openai.com/docs/assistants/tools/code-interpreter
5. https://github.com/Significant-Gravitas/Auto-GPT

to exploratory workflows or tasks with undefined dependencies. Additionally, it lacks direct integration of domain-specific insights, such as clinical knowledge for healthcare datasets.

There are also additional LLM-based data cleaning tools or other agent based systems which have been proposed in the literature. In Table 7 we highlight these and why they are not suitable for comparison with CliMB-DC.

Table 7: Summary of alternative data cleaning frameworks or other agent-based frameworks and reasons for not comparing with CliMB-DC.

| Framework | Reason Not Used |
|---|---|
| CleanAgent (Qi et al., 2024) | Designed specifically for dataset cleaning, not a complete pipeline for data modeling. Reported metrics focus on cell-level matching rates, making it unsuitable as a baseline. |
| MatPlotAgent (Yang et al., 2024) | Primarily designed for data visualization, irrelevant to our data modeling approach. |
| WaitGPT (Xie et al., 2024) | Intended for transforming code into user interfaces for verifying execution. Goals are fundamentally different from ours. |
| SEED (Chen et al., 2023) | Focuses on domain-specific data curation (imputation, annotation, discovery), but goals do not align with Climb-DC. No code released. |
| AutoM3L (Luo et al., 2024) | Designed for multi-modal data analysis. Code is not available, making comparison infeasible. |
| HuggingGPT (Shen et al., 2023) | Geared towards general AI tasks, not designed for tabular data analysis or modeling. |
| MLCopilot (Zhang et al., 2024) | Lacks a planning mechanism and requires user-specified actions, functioning more as a copilot than an autonomous agent. |
| SELA (Chi et al., 2024) | Requires significant pre-processing and is not adaptable to all use cases (e.g., multiple data files). Limited applicability. |
| AutoML-Agent (Trirat et al., 2024) | Comparison relevant, but requires GPUs to run. Uploading our data to GPU-supported servers is not feasible. |
| AutoKaggle (Li et al., 2025) | Designed for Kaggle competitions, requiring train/test splits and submission templates, incompatible with our goals. |
| LAMBDA (Sun et al., 2025) | We include a comparison and show Climb-DC outperforms it; struggles with issues like label leakage. |

## A.2 Limitations Across Approaches

Despite their individual strengths, these systems share several overarching limitations that hinder their applicability to real-world, data-centric workflows:

- **Static Pipeline Architectures:** Most interpreters rely on predefined templates or fixed task hierarchies, which restrict their ability to adapt to evolving requirements. For example, AutoGPT and DS-Agent struggle with workflows where task dependencies are contingent on intermediate results.

- **Insufficient Data Reasoning:** While these tools excel at executing predefined workflows, they lack higher-level data-centric reasoning capabilities, such as identifying feature correlations, addressing data drift, or diagnosing feature leakage. For instance, GPT-Code Interpreter and OpenHands fail to contextualize data preprocessing steps to account for domain-specific nuances.

- **Healthcare-Specific Challenges:** Healthcare datasets present unique challenges, including heterogeneity, noise, and biases. Generic preprocessing approaches risk introducing errors or obscuring critical clinical information. For example, extreme lab values might appear as statistical outliers but could signify a critical medical condition. Current systems fail to incorporate the domain expertise required to navigate such complexities.

- **Limited Adaptability to Data Evolution:** Real-world datasets often exhibit evolving distributions, feature sets, or objectives. Most interpreters, including BLADE and Data Interpreter, are designed for static workflows and do not account for the dynamic nature of these datasets.

- **Lack of Control in Open-Ended Scenarios:** Systems with open-ended prompting, such as AutoGPT and DS-Agent, can generate uncontrolled outputs when used by non-experts. This is particularly problematic in sensitive domains like healthcare, where errors can lead to significant consequences.

### A.3 Terminology differences between Machine Learning (ML) and Biostatistics/Epidemiology

Table 8 demonstrates the different terminologies between communities that it is useful for a co-pilot to handle.

Table 8: Comparison of terminology in Machine Learning (ML) and Biostatistics/Epidemiology, which is considered in the design of CliMB-DC.

| Machine Learning (ML) | Biostatistics/Epidemiology |
| --- | --- |
| Model/Algorithm | Statistical/Predictive Model |
| Features | Covariates/Covariables |
| Targets | Outcomes/Endpoints |
| Training | Model Fitting/Estimation |
| Test Set | Validation Data |
| Overfitting | Overparameterization |
| Hyperparameters | Tuning Parameters |
| Performance Metrics | Goodness-of-Fit Measures |
| Cross-Validation | Internal Validation |
| Bias-Variance Tradeoff | Model Complexity |
| Generalization | External Validity |
| Feature Selection | Variable Selection |

## Appendix B. Dataset Descriptions

**Dataset availability:** The *Lung Cancer Dataset*, *Primary Biliary Cholangitis (PBC) Dataset* and *CUTRACT* datasets are confidential medical datasets and cannot be released. The underlying data for the *SEER* dataset may be requested from SEER (see https://seer.cancer.gov/data/). The *SFLD* dataset used in Appendix E.4 can be obtained from Freddie Mac (2025).

### B.1 Lung Cancer Dataset

The dataset consists of **216714** records, capturing baseline and follow-up data related to lung cancer risk factors, smoking history, and demographic attributes. It includes **31 features**, broadly categorized as follows:

- **Demographic and Administrative:** This category includes age at recruitment, sex, ethnicity, and highest qualifications attained.

- **Smoking History:** Features include the number of cigarettes smoked per day, age at which smoking started and stopped, smoking duration, years since quitting, and pack-years (a cumulative measure of smoking exposure).

- **Respiratory and Comorbid Conditions:** This includes self-reported history of respiratory diseases such as asbestosis, pneumonia, chronic obstructive pulmonary disease (COPD), emphysema, chronic bronchitis, asthma, and allergic conditions (eczema, allergic rhinitis, hay fever).

- **Cancer History:** The dataset captures both personal and family history of lung cancer, including lung cancer diagnoses in parents (mother and father) and siblings, as well as the number of self-reported cancers and prior personal history of cancer.

- **Occupational and Environmental Exposure:** Presence of asbestos exposure is recorded as a binary indicator.

- **Lung Cancer Outcomes and Time-to-Event Data:** The dataset includes indicators for lung cancer diagnosis and related outcomes, along with survival-related features such as time to lung cancer diagnosis, time to death from lung cancer, and time to event development.

  ***Task.*** Lung cancer risk prediction and survival analysis

### B.2 Primary Biliary Cholangitis (PBC) Dataset

The dataset consists of **43,834** records across 2181 patients, capturing baseline and follow-up data for individuals diagnosed with **Primary Biliary Cholangitis (PBC)**. It includes **33 features**, broadly categorized as follows:

- **Demographic and Administrative:** This category includes patient ID, sex, age, visit type, and time-related variables, which provide essential context for each recorded observation.

- **Clinical Outcomes:** Features in this category capture event status, survival status, and liver transplantation (LT) status, allowing for disease progression analysis.

- **Clinical Complications:** These features focus on manifestations of liver dysfunction, including decompensation (Decomp), variceal hemorrhage (VH), ascites, and hepatic encephalopathy (HE).

- **Treatment Variables:** This category records the use of Ursodeoxycholic Acid (UDCA), Obeticholic Acid (OCA), and Bezafibrate (BZF), which are commonly used interventions in PBC management.

- **Laboratory Measurements:** Example biomarkers include Albumin, Bilirubin, ALP, ALT, Platelets, Hemoglobin, White Cell Count, Urea, Creatinine, Sodium, Potassium, IgM, IgG, IgA

- **Comorbidity Assessment:** The Charlson Comorbidity Index (CCI) score is included as a prognostic measure for patient risk stratification.

Note the PBC dataset has repeat measurements for each patient that need to be aggregated before modelling.

*Task.* Time-to-event modeling and survival analysis.

## B.3 Prostate Cancer Prediction: SEER and CUTRACT datasets

This task focuses on **10-year prostate cancer mortality prediction** using two datasets: **SEER** (Surveillance, Epidemiology, and End Results) from the **United States** and **CU-TRACT** from the **United Kingdom**. The goal is to assess how models trained in one country generalize when deployed in another, particularly in handling distribution shifts across different healthcare systems.

Both datasets are balanced to 2000 patient records each, with **10 features** related to patient demographics, cancer severity, treatment, and outcomes. The features are categorized as follows:

- **Demographic and Clinical Characteristics:** Age at diagnosis (`age`), baseline Prostate-Specific Antigen (`psa`), and presence of comorbidities (`comorbidities`).

- **Treatment Variables:** Indicators of different treatment modalities, including conservative management (`treatment_CM`), primary hormone therapy (`treatment_Primary hormone therapy`), radical radiotherapy (`treatment_Radical Therapy-RDx`), and radical surgery (`treatment_Radical therapy-Sx`).

- **Tumor Characteristics:** Cancer grade (`grade`) and stage (`stage`), which define disease severity.

- **Outcome Variable:** The binary label `mortCancer` represents **10-year mortality due to prostate cancer**, serving as the prediction target.

*Task.* 10-year prostate cancer prediction (binary classification). Given the multi-country for train and test it serves as an example of cross-national deployment of prediction models and their robustness to distribution shift.

## Appendix C. Interaction Examples

In this section, we present execution logs from both the baselines and CliMB-DC to illustrate two key points: (1) why the baselines fail to address data issues effectively, and (2) how CliMB-DC successfully tackles these challenges in healthcare. We first show examples of baselines, i.e. Data-Interpreter and OpenHands, and then provide the corresponding successful examples of CliMB-DC.

### C.1 Data-Interpreter

#### C.1.1 EXAMPLE: INCORRECT METRIC USED

In the following example (lung cancer dataset), we observe a failure case with Data-Interpreter, where it used the ROC-AUC score as a proxy for the C-index score. This issue arises due to the absence of Python packages for survival analysis. Since Data-Interpreter lacks user interaction capabilities, the system failed to detect and address this error.

---

**Summary of work up to this point:** The datasets have been successfully loaded and initially preprocessed. However, despite multiple attempts to utilize Python packages for survival analysis, such as `lifelines` and `sksurv`, the installation process has not been triggered.

...

**Generated code:**

```
1   import xgboost as xgb
2   from sklearn.model_selection import train_test_split
3   from sklearn.metrics import roc_auc_score
4   import numpy as np
5   # Prepare the data for survival analysis
6   X_train = train_df_copy.drop(columns=['time_to_event_develop', '
        lung_cancer'])
7   y_train = train_df_copy[['time_to_event_develop', 'lung_cancer']]
8   X_eval = eval_df_copy.drop(columns=['time_to_event_develop', '
        lung_cancer'])
9   y_eval = eval_df_copy[['time_to_event_develop', 'lung_cancer']]
10  # Convert the target to a structured array for xgboost
11  y_train_structured = np.array([(time, event) for time, event in zip(
        y_train['time_to_event_develop'], y_train['lung_cancer'])],
12                                 dtype=[('time', 'float'), ('event', '
                                     bool')])
13  y_eval_structured = np.array([(time, event) for time, event in zip(
        y_eval['time_to_event_develop'], y_eval['lung_cancer'])],
14                                 dtype=[('time', 'float'), ('event', 'bool
                                     ')])
15  # Create DMatrix for xgboost
16  dtrain = xgb.DMatrix(X_train, label=y_train['lung_cancer'])
17  deval = xgb.DMatrix(X_eval, label=y_eval['lung_cancer'])
18  # Set up parameters for xgboost
19  params = {
20      'objective': 'binary:logistic',
21      'eval_metric': 'auc',
22      'max_depth': 3,
23      'eta': 0.1,
```

```
24          'subsample': 0.8,
25          'colsample_bytree': 0.8,
26          'seed': 42
27      }
28      # Train the xgboost model
29      bst = xgb.train(params, dtrain, num_boost_round=100)
30      # Predict the risk scores for the evaluation set
31      risk_scores = bst.predict(deval)
32      # Calculate the concordance index
33      c_index = roc_auc_score(y_eval['lung_cancer'], risk_scores)
34      print(f"C-index: {c_index:.4f}")
```

**Code execution output:**
Code execution finished successfully ✓

```
1   C-index: 0.5337
```

**Reason behind this.**  Data-Interpreter lacks the ability to interact with users, making it difficult to trigger the installation of required Python packages when they are missing. Additionally, since survival analysis is not as common as standard machine learning tasks like classification and regression, the system fails to recognize how to compute the C-index score. As a result, it incorrectly applies an inappropriate metric without detecting the issue.

### C.1.2 Example: Fail to setup survival analysis

In the following example (lung cancer dataset), we observe a failure case with Data-Interpreter, where it *failed to setup the survival analysis*. The system "auto-degraded" to XGBoost and failed to address this error.

---

**Summary of work up to this point:** The datasets have been successfully loaded and initially preprocessed. Python packages for survival analysis, such as `lifelines` and `sksurv`, are *manually* installed. However, it encountered "Convergence error" when fitting `CoxPHFitter` due to the high collinearity.

. . .

**Generated code:**

```
1    from xgboost import XGBRegressor
2    from lifelines.utils import concordance_index
3    import pandas as pd
4    # Prepare the data
5    X_train = train_data_var.drop(columns=['time_to_event_develop', '
         lung_cancer'])
6    y_train = train_data_var[['time_to_event_develop', 'lung_cancer']]
7    X_eval = eval_data_var.drop(columns=['time_to_event_develop', '
         lung_cancer'])
8    y_eval = eval_data_var[['time_to_event_develop', 'lung_cancer']]
9    # Train the XGBoost model
10   xgb_model = XGBRegressor(objective='survival:cox', n_estimators=100,
         learning_rate=0.1, max_depth=3, subsample=0.8, colsample_bytree
         =0.8)
11   xgb_model.fit(X_train, y_train['time_to_event_develop'])
```

```
12    # Predict risk scores
13    eval_data_combined['risk_score'] = xgb_model.predict(X_eval)
14
15    # Calculate the C-index
16    c_index = concordance_index(y_eval['time_to_event_develop'], -
          eval_data_combined['risk_score'], y_eval['lung_cancer'])
17    print(f"C-index: {c_index}")
```

> **Code execution output:**
> Code execution finished successfully ✓
> ```
> 1  C-index: 0.5000
> ```

**Reason behind this.** Data-Interpreter lacks both data-centric tools and sufficient knowledge of data preprocessing to address collinearity effectively. As a result, even when provided with the appropriate survival analysis package (`lifelines`), it repeatedly encounters "Convergence errors." Eventually, it abandons the use of `lifelines` and reverts to using XGBoost.

### C.1.3 EXAMPLE: FAIL TO ADDRESS LABEL LEAKAGE

In the following example (PBC dataset), we observe a failure case with Data-Interpreter, where it *failed to address the data aggregation as well as the label leakage*. The system did not recognize such issues.

---

**Summary of work up to this point:** The datasets have been successfully loaded and initially preprocessed. Python packages for survival analysis, such as `lifelines` and `sksurv`, are *manually* installed.

**Generated code:**

```
1    # Evaluate the model using 5-fold cross-validation and report the C-
         index
2    import pandas as pd
3    from metagpt.tools.libs.data_preprocess import FillMissingValue
4    from sklearn.model_selection import KFold
5    from sklearn.preprocessing import LabelEncoder, StandardScaler
6    from lifelines import CoxPHFitter
7    from lifelines.utils import concordance_index
8    import numpy as np
9    # Load and preprocess the data
10   data_path = '/shared/share_mala/jiashuo/MetaGPT/data/di_dataset/
         copilot/PBCData.csv'
11   df = pd.read_csv(data_path, encoding='ISO-8859-1')
12
13   # Fill missing values
14   df_copy = df.copy()
15   numeric_features = df_copy.select_dtypes(include=['float64', 'int64'])
         .columns.tolist()
16   fill_missing_numeric = FillMissingValue(features=numeric_features,
         strategy='mean')
17   df_copy = fill_missing_numeric.fit_transform(df_copy)
18   categorical_features = df_copy.select_dtypes(include=['object']).
         columns.tolist()
```

```
19    fill_missing_categorical = FillMissingValue(features=
          categorical_features, strategy='most_frequent')
20    df_copy = fill_missing_categorical.fit_transform(df_copy)
21    # Encode categorical features
22    label_encoders = {}
23    for col in ['Visit_Type', 'Sex', 'Event_Category', 'Alive_status', 'LT
          ', 'Decomp', 'VH', 'Ascites', 'HE']:
24        le = LabelEncoder()
25        df_copy[col] = le.fit_transform(df_copy[col].astype(str))
26        label_encoders[col] = le
27    # Fill remaining missing values
28    df_copy.fillna(df_copy.mean(numeric_only=True), inplace=True)
29    for col in ['Visit_Type', 'Sex', 'Event_Category', 'Alive_status', 'LT
          ', 'Decomp', 'VH', 'Ascites', 'HE']:
30        df_copy[col].fillna(df_copy[col].mode()[0], inplace=True)
31
32    # Drop low variance columns
33    low_variance_cols = [col for col in df_copy.columns if df_copy[col].
          var() < 1e-5]
34    df_copy.drop(columns=low_variance_cols, inplace=True)
35
36    # Prepare features and target
37    X = df_copy.drop(columns=['Event_status', 'Years_to_event'])
38    y = df_copy[['Event_status', 'Years_to_event']]
39    # Standardize features
40    scaler = StandardScaler()
41    X_scaled = scaler.fit_transform(X)
42    X_scaled = pd.DataFrame(X_scaled, columns=X.columns)
43
44    # Perform 5-fold cross-validation
45    kf = KFold(n_splits=5, shuffle=True, random_state=42)
46    c_indices = []
47
48    for train_index, test_index in kf.split(X_scaled):
49        X_train, X_test = X_scaled.iloc[train_index], X_scaled.iloc[
              test_index]
50        y_train, y_test = y.iloc[train_index], y.iloc[test_index]
51
52        train_data = X_train.copy()
53        train_data['Event_status'] = y_train['Event_status']
54        train_data['Years_to_event'] = y_train['Years_to_event']
55        test_data = X_test.copy()
56        test_data['Event_status'] = y_test['Event_status']
57        test_data['Years_to_event'] = y_test['Years_to_event']
58
59        cph = CoxPHFitter(penalizer=0.1)
60        cph.fit(train_data, duration_col='Years_to_event', event_col='
              Event_status')
61
62        test_predictions = cph.predict_partial_hazard(test_data)
63        c_index = concordance_index(test_data['Years_to_event'], -
              test_predictions, test_data['Event_status'])
64        c_indices.append(c_index)
65    mean_c_index = np.mean(c_indices)
66    print(f'Mean C-index from 5-fold cross-validation: {mean_c_index}')
```

---

**Code execution output:**
Code execution finished successfully ✓

```
1   Mean C-index from 5-fold cross-validation: 0.7750170186750802
```

---

**Reason behind this.** Although Data-Interpreter successfully fitted a Cox regression model for survival analysis, it failed to address key data issues in the PBC dataset. Specifically, the PBC data contains multiple observations per patient ID, requiring aggregation prior to analysis. Additionally, besides the "Event_status" and "Years_to_event" features, there are "Event_Category" and "Alive_status" features that are highly correlated with the target variable, leading to label leakage. However, Data-Interpreter did not account for these issues, resulting in flawed analysis.

## C.2 OpenHands

In the following example (lung cancer dataset), we observe a failure case with OpenHands, where it (1) *failed to appropriately preprocess the data* and (2) *failed to deal with label leakage*. The system did not recognize or deal with such issues.

---

**Summary of work up to this point:** The datasets have been successfully loaded and initially preprocessed.

```python
1   import pandas as pd
2   from lifelines import CoxPHFitter
3   from lifelines.utils import concordance_index
4
5   # Load the training and testing datasets
6   train_data = pd.read_csv('/workspace/copilot/lung_train.csv')
7   test_data = pd.read_csv('/workspace/copilot/lung_test.csv')
8
9   # Prepare the data for survival analysis
10  X_train = train_data.drop(columns=['time_to_event_develop', 'lung_cancer'
        ])
11  y_train = train_data[['time_to_event_develop', 'lung_cancer']]
12
13  # Fit the Cox Proportional Hazards model
14  cph = CoxPHFitter()
15  cph.fit(y_train, duration_col='time_to_event_develop', event_col='
        lung_cancer')
16
17  # Prepare the test data
18  X_test = test_data.drop(columns=['time_to_event_develop', 'lung_cancer'])
19  y_test = test_data[['time_to_event_develop', 'lung_cancer']]
20
21  # Predict the risk scores for the test data
22  risk_scores = cph.predict_partial_hazard(X_test)
23
24  # Calculate the C-index
25  c_index = concordance_index(y_test['time_to_event_develop'], -risk_scores
        , y_test['lung_cancer'])
26
27  c_index
```

```
Code execution output:
Code execution finished successfully ✓
1 np.float64(0.5)
```

**Assistant:** The survival prediction model for lung cancer has been built using the Cox Proportional Hazards model. The C-index reported on the test dataset is 0.5. However, the model summary is empty, which may indicate that the model did not find any significant predictors in the training data. If you need further analysis or adjustments, please let me know!

**Reason behind this.** The prediction model developed by OpenHands demonstrates subpar performance. This is primarily due to the lack of any data preprocessing, resulting in an ineffective and meaningless survival model. Besides, OpenHands did not recognize the label leakage problem caused by feature "time_to_death_lung_cancer" and "time_to_lung_cancer". Moreover, OpenHands failed to identify the label leakage issue caused by the features "time_to_death_lung_cancer" and "time_to_lung_cancer". The failure scenarios closely resemble those observed with Data-Interpreter.

### C.3 Successful Cases with CliMB-DC

Here, we demonstrate how CliMB-DC effectively addresses the previously discussed failure cases, using the lung cancer dataset as an example.

#### C.3.1 Example: Address label leakage

This example demonstrates how CliMB-DC effectively addresses label leakage issues via advanced reasoning and human interaction.

---

**Summary of Progress to Date:** The datasets have been successfully loaded and appropriately preprocessed, including steps such as imputation, normalization, and other necessary adjustments. Before initiating experiments, CliMB-DC ensures alignment with the user by confirming the problem setup, while also checking for potential label leakage.
. . .

**Assistant:** I will now consult the message history to check the meaning and details of each of these columns and identify any potential data leakage.

**Suspected Data Leakage Columns:**

- "time_to_death_lung_cancer": This column is likely to reveal the target variable "lung_cancer".

- "time_to_lung_cancer": This column is likely to reveal the target variable "lung_cancer".

Data leakage is a problem because it can lead to overly optimistic results and a model that cannot be used in practice. By including information that would not be available at the time of prediction, the model can learn patterns that are not generalizable to new data.

**Assistant:** Do you agree with the identified columns for potential data leakage? If there are any other columns you suspect might cause data leakage, please let me know.

**User:** Yes, those two are data leakage. Remove them please

**Assistant:** I will now generate the code to remove the identified columns ("time_to_death_lung_cancer" and "time_to_lung_cancer") from both the training and test datasets.

**Generated code:**

```python
import pandas as pd

# Load the training dataset
train_df = pd.read_csv("lung_train.csv")

# Drop the identified columns
train_df = train_df.drop(columns=["time_to_death_lung_cancer", "
    time_to_lung_cancer"])

# Save the modified training dataset
train_df.to_csv("lung_train_no_leakage.csv", index=False)

# Load the test dataset
test_df = pd.read_csv("lung_test.csv")

# Drop the identified columns
test_df = test_df.drop(columns=["time_to_death_lung_cancer", "
    time_to_lung_cancer"])

# Save the modified test dataset
test_df.to_csv("lung_test_no_leakage.csv", index=False)

print("Columns 'time_to_death_lung_cancer' and 'time_to_lung_cancer'
    have been removed from both datasets.")
```

**Code execution output:**
Code execution finished successfully ✓

```
Columns ''time_to_death_lung_cancer'' and ''time_to_lung_cancer''
    have been removed from both datasets.
```

. . .

**Summary of Subsequent Work:** CliMB-DC then went on to identifying irrelevant features to remove. An illustrative example is provided below.

### C.3.2 EXAMPLE: IDENTIFY IRRELEVANT FEATURES

This example demonstrates how CliMB-DC identified irrelevant features and interacted with the user to remove them.

---

**Summary of Progress to Date:** The datasets have been appropriately preprocessed and the label leakage issue has been addressed based on expert human feedback. Moving forward, CliMB-DC focuses on identifying and addressing irrelevant features.

. . .

> **Assistant:** I will now check if there are any meaningless/irrelevant columns still left in the dataset.
>
> **Suspected Irrelevant Columns:**
>
> - "index": This column is likely an identifier and does not contain any useful information for the analysis.
>
> Including irrelevant columns can lead to overfitting and misleading feature importance. It is important to remove them before continuing with the machine learning study.

> **Assistant:** Do you agree with removing the "index" column? Are there any other columns you suspect might be irrelevant?

> **User:** Yes, remove "index" column only.

> **Assistant:** I will now generate the code to remove the "index" column from both the training and test datasets.
> **Generated code:**
>
> ```python
> import pandas as pd
>
> # Load the training dataset
> train_df = pd.read_csv("lung_train_no_leakage.csv")
>
> # Drop the identified irrelevant column
> train_df = train_df.drop(columns=["index"])
>
> # Save the modified training dataset
> train_df.to_csv("lung_train_no_leakage_no_index.csv", index=False)
>
> # Load the test dataset
> test_df = pd.read_csv("lung_test_no_leakage.csv")
>
> # Drop the identified irrelevant column
> ```

```
16  test_df = test_df.drop(columns=["index"])
17
18  # Save the modified test dataset
19  test_df.to_csv("lung_test_no_leakage_no_index.csv", index=False)
20
21  print("Column 'index' has been removed from both datasets.")
```

**Code execution output:**
Code execution finished successfully ✓

```
1  Column ''index'' has been removed from both datasets.
```

**Assistant:** The index column has been removed from both the training and test datasets. The modified datasets have been saved as lung_train_no_leakage_no_index.csv and lung_test_no_leakage_no_index.csv.

**Feature Columns Left**: Here are the feature columns that are left in the dataset:

- age_at_recruitment

- sex

- body_mass_index

- n_cig_per_day

- ...

Are you happy to use all of these features in the machine learning study?

**User:** Yes

. . .

**Summary of Subsequent Work:** CliMB-DC then went on to utilizing domain-specific models tailored for survival analysis. An illustrative example is provided below.

C.3.3 Example: Setup domain-specific model & metric

Following the above examples, this example shows how CliMB-DC leveraged domain-specific tools to build survival models effectively.

**Summary of Progress to Date:** The datasets have been appropriately preprocessed and the label leakage issue has been addressed based on expert human feedback. Irrelevant features are removed.

. . .

**Assistant:** Tool `autoprognosis_survival_train_test` completed successfully ✓
**Tool logs:**

```
 1  Trying models:
 2  [
 3    "cox_ph",
 4    "deephit"
 5  ]
 6  Running the survival analysis study, this may take several minutes...
 7  AutoPrognosis 2.0: Duration of this iteration: 15.6s
 8  AutoPrognosis 2.0: Investigating model: data_cleanup->cox_ph
 9  AutoPrognosis 2.0: C-index achieved: 0.917
10  AutoPrognosis 2.0: Brier score achieved: 0.003
11  AutoPrognosis 2.0: Duration of this iteration: 4.1s
12  AutoPrognosis 2.0: Investigating model: nop->feature_normalizer->
       data_cleanup->cox_ph
13  AutoPrognosis 2.0: C-index achieved: 0.896
14  AutoPrognosis 2.0: Brier score achieved: 0.003
15  AutoPrognosis 2.0: Duration of this iteration: 3.7s
16  AutoPrognosis 2.0: Investigating model: nop->normal_transform->
       data_cleanup->cox_ph
17  AutoPrognosis 2.0: C-index achieved: 0.892
18  AutoPrognosis 2.0: Brier score achieved: 0.003
19  AutoPrognosis 2.0: Duration of this iteration: 3.4s
20  AutoPrognosis 2.0: Investigating model: data_cleanup->cox_ph
21  AutoPrognosis 2.0: C-index achieved: 0.898
22  AutoPrognosis 2.0: Brier score achieved: 0.007
23  AutoPrognosis 2.0: Duration of this iteration: 3.7s
24  AutoPrognosis 2.0: Investigating model: nop->feature_normalizer->
       data_cleanup->cox_ph
25  AutoPrognosis 2.0: C-index achieved: 0.862
26  AutoPrognosis 2.0: Brier score achieved: 0.007
27  AutoPrognosis 2.0: Duration of this iteration: 3.4s
28  AutoPrognosis 2.0: Investigating model: nop->normal_transform->
       data_cleanup->cox_ph
29  AutoPrognosis 2.0: C-index achieved: 0.876
30  AutoPrognosis 2.0: Brier score achieved: 0.007
31  AutoPrognosis 2.0: Duration of this iteration: 2.9s
32  AutoPrognosis 2.0: Investigating model: data_cleanup->cox_ph
33  AutoPrognosis 2.0: C-index achieved: 0.843
34  AutoPrognosis 2.0: Brier score achieved: 0.012
35  AutoPrognosis 2.0: Duration of this iteration: 3.2s
36  AutoPrognosis 2.0: Investigating model: nop->feature_normalizer->
       data_cleanup->cox_ph
37  AutoPrognosis 2.0: C-index achieved: 0.803
38  AutoPrognosis 2.0: Brier score achieved: 0.012
39  AutoPrognosis 2.0: Duration of this iteration: 3.0s
40  AutoPrognosis 2.0: Investigating model: nop->normal_transform->
       data_cleanup->cox_ph
41  AutoPrognosis 2.0: C-index achieved: 0.830
42  AutoPrognosis 2.0: Brier score achieved: 0.012
43  AutoPrognosis 2.0: Duration of this iteration: 7.6s
44  AutoPrognosis 2.0: Duration of this iteration: 7.6s
45  AutoPrognosis 2.0: Investigating model: ['1.0 * data_cleanup->cox_ph']
```

```
46  AutoPrognosis 2.0: C-index achieved: 0.952 +/- 0.0
47  AutoPrognosis 2.0: Brier score achieved: 0.003 +/- 0.0
48  AutoPrognosis 2.0: Investigating model:
49  ['1.0 * data_cleanup->cox_ph']
50  AutoPrognosis 2.0: C-index achieved: 0.952 +/- 0.0
51  AutoPrognosis 2.0: Brier score achieved: 0.003 +/- 0.0
52  AutoPrognosis 2.0: Duration of this iteration: 7.6s
53  AutoPrognosis 2.0: Duration of this iteration: 7.6s
54  AutoPrognosis 2.0: Investigating model: ['1.0 * data_cleanup->cox_ph']
55  AutoPrognosis 2.0: C-index achieved: 0.936 +/- 0.0
56  AutoPrognosis 2.0: Brier score achieved: 0.007 +/- 0.0
57  AutoPrognosis 2.0: Investigating model:
58  ['1.0 * data_cleanup->cox_ph']
59  AutoPrognosis 2.0: C-index achieved: 0.936 +/- 0.0
60  AutoPrognosis 2.0: Brier score achieved: 0.007 +/- 0.0
61  AutoPrognosis 2.0: Duration of this iteration: 7.1s
62  AutoPrognosis 2.0: Duration of this iteration: 7.1s
63  AutoPrognosis 2.0: Investigating model: ['1.0 * data_cleanup->cox_ph']
64  AutoPrognosis 2.0: C-index achieved: 0.87 +/- 0.0
65  AutoPrognosis 2.0: Brier score achieved: 0.012 +/- 0.0
66  AutoPrognosis 2.0: Investigating model:
67  ['1.0 * data_cleanup->cox_ph']
68  AutoPrognosis 2.0: C-index achieved: 0.87 +/- 0.0
69  AutoPrognosis 2.0: Brier score achieved: 0.012 +/- 0.0
70  AutoPrognosis 2.0: Duration of this iteration: 609.3s
71  AutoPrognosis 2.0: Duration of this iteration: 609.3s
72  AutoPrognosis 2.0: Investigating model: ["1.0 * data_cleanup->cox_ph
        ({'cox_ph': {}})", "1.0 * data_cleanup->cox_ph({'cox_ph': {}})", "
        1.0 * data_cleanup->cox_ph({'cox_ph': {}})"]
73  AutoPrognosis 2.0: C-index achieved: 0.903
74  AutoPrognosis 2.0: Brier score achieved: 0.007
75  AutoPrognosis 2.0: Investigating model:
76  AutoPrognosis 2.0: Study finished.
77  ["1.0 * data_cleanup->cox_ph({'cox_ph': {}})", "1.0 * data_cleanup->
        cox_ph({'cox_ph': {}})", "1.0 * data_cleanup->cox_ph({'cox_ph':
        {}})"]
78  AutoPrognosis 2.0: C-index achieved: 0.903
79  AutoPrognosis 2.0: Brier score achieved: 0.007
80  AutoPrognosis 2.0: Study finished.
81  Survival analysis study completed!
82  Final model summary:
83  Metrics (train data): "c_index": "0.925 +/- 0.029",
84  Metrics (test data): "c_index": "0.848 +/- 0.005"
```

### C.4 Example: Model Evaluation & Feature Importance

Based on the results, CliMB-DC will provide feature importance explanations for the users to better understand the prediction model, ensuring the interpretability in healthcare.

---

**Summary of Progress to Date:** CliMB-DC has fitted a survival model for the given dataset and the task.

. . .

**Assistant:** Would you like to see feature importance plots for your survival analysis task?

**User:** Yes, please

**Assistant:** Tool `permutation_explainer` completed successfully ✓
**Tool logs:**

```
1  Loading the data...
2  Data loaded with shape: 4800 rows and 28 columns.
3  Loading model from file: path/model_all.p
4  Running the permutation explainer, this can take a while...
5
6         This tool takes approximately 1 minute per column. If this is
              too slow, consider reducing the number of columns.
7         Reducing the number of column can be done with the feature
              selection tool. You can cancel this tool with the
8         'Restart from last reasoning step' button. Then use the
              feature selection tool to reduce the number of columns.
9  Permutation explainer completed!
10                          importances_mean   importances_std
11 age_at_recruitment             0.088055          0.019220
12 copd_emphysema_bronchitis      0.018188          0.004543
13 packyears                      0.018143          0.007201
14 pneumonia_self_reported        0.017176          0.000252
15 age_started_smoking            0.015429          0.002090
16 personal_hx_cancer             0.010980          0.003070
17 ...
```

**Assistant:** The feature importance plot is here[a]:

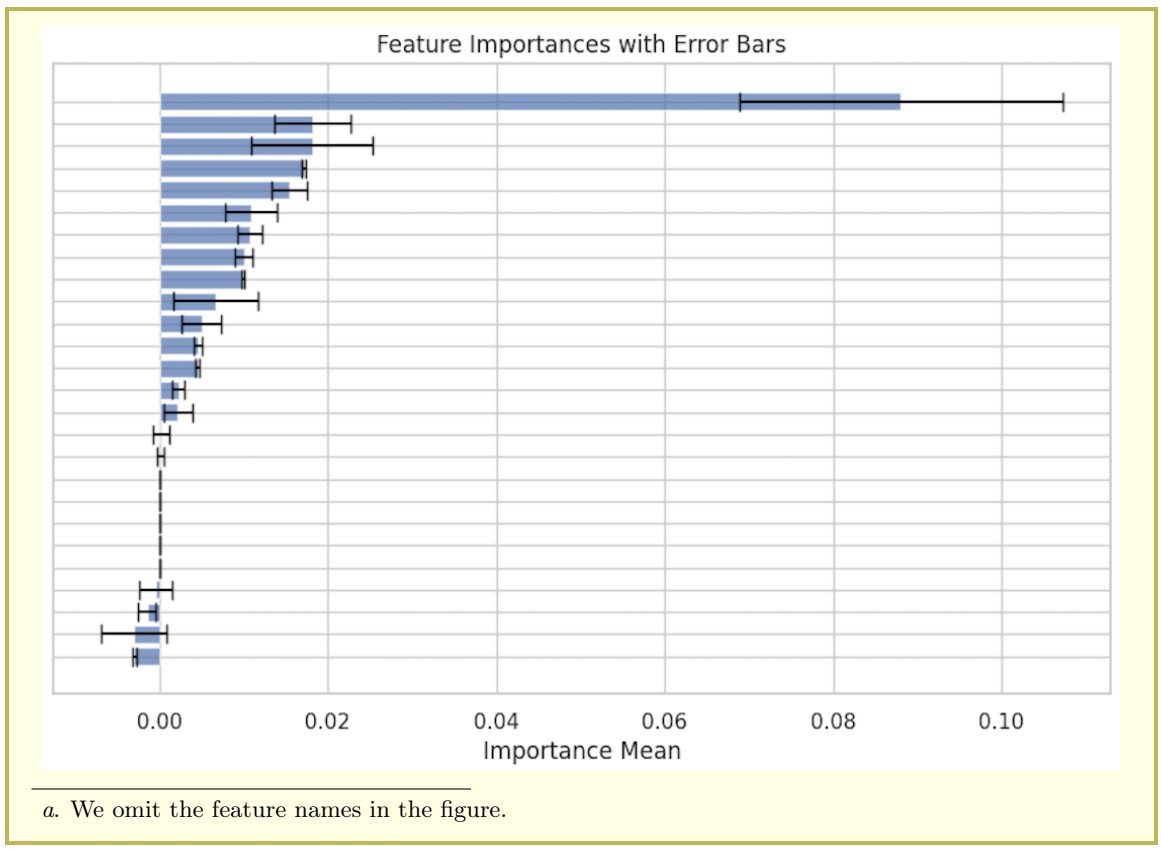

*a.* We omit the feature names in the figure.

## Appendix D. More CliMB-DC Interaction Examples

In this section, we present more examples highlighting the advantages of CliMB-DC in executing effective data preprocessing.

### D.1 Example: Data Exploration

For the input data, CliMB-DC provides users with a detailed data analysis as well as visualization using tools `EDA`, `descriptive statistics`, and `lifelines.KaplanMeierFitter`.

---

**Summary of Progress to Date**: The data file has been uploaded.

. . .

---

**Assistant:** Tool `EDA` completed successfully ✓

**Tool logs:**

```
1  Getting dataset basic info...
2  Getting descriptive statistics for numerical features...
3  Getting detailed information on categorical variables...
4  Performing missing values analysis...
5  Identified numeric columns that should be considered categorical:
```

```
6  ['asbestosis_self_reported', 'pneumonia_self_reported', '
       chronic_obstructive_airways_disease_copd_self_reported', '
       emphysema_chronic_bronchitis', 'copd_emphysema_bronchitis', '
       emphysema', 'asthma', 'eczema_allergicrhinitis_hayfever', '
       number_of_self_reported_cancers', 'personal_hx_cancer', '
       family_hx_lung_cancer_mother', 'family_hx_lung_cancer_father', '
       family_hx_lung_cancer_siblings', 'asbestos', 'lung_cancer']
7  Performing correlation analysis...
8  Performing potential outliers identification...
9  Performing duplicate records analysis...
```

**Tool output:**

```
1  Dataset Shape: 4800 rows and 31 columns
2  Column Names and Types:
3  index                                             int64
4  age_at_recruitment                                float64
5  sex                                               object
6  body_mass_index                                   float64
7  n_cig_per_day                                     float64
8  age_started_smoking                               float64
9  age_stopped_smoking                               float64
10 ...
11
12 Descriptive Statistics for Numerical Features:
13              index    age_at_recruitment    body_mass_index ...
14 count   4800.000000          4800.000000        4800.000000 ...
15 mean    87406.250417           57.417163          27.676107 ...
16 std     49908.726761            8.084722           4.621542 ...
17 min       129.000000           40.000000          16.024943 ...
18 25%     44130.250000           51.167603          24.473217 ...
19 50%     88896.000000           59.135000          27.116986 ...
20 75%    130874.500000           63.719537          30.425950 ...
21 max    173314.000000           72.000000          53.565422 ...
22 skew       -0.033166           -0.447521           0.579946 ...
23 kurt       -1.179482           -0.848562           0.532072 ...
24
25 Identified numeric value columns that should most likely be considered
       categoricals:
26 ['asbestosis_self_reported', 'pneumonia_self_reported', 'asbestos', '
       lung_cancer', ...].
27 This is done by checking whether the column contains only integers and
       has a low number of unique values (<20 or <5% of total examples).
28
29 Detailed Information on Categorical Variables:
30 smoking_status - Unique Values: 2
31 Top 5 Values:
32 smoking_status
33 Previous    3603
34 Current     1197
35 ...
36
37 Missing Values Analysis:
38 No missing values found.
39
```

```
40  Correlation  Analysis:  ...
41
42  Outlier  Identification  for  Numerical  Features:
43  index  -  Outliers  Count:  0
44  [Lower  Bound:  -1.08e+05,  Upper  Bound:  2.83e+05]
45  ...
46
47  Duplicate  Records:  0
```

**Assistant:** Here is a correlation gram showing the correlation between features[a]:

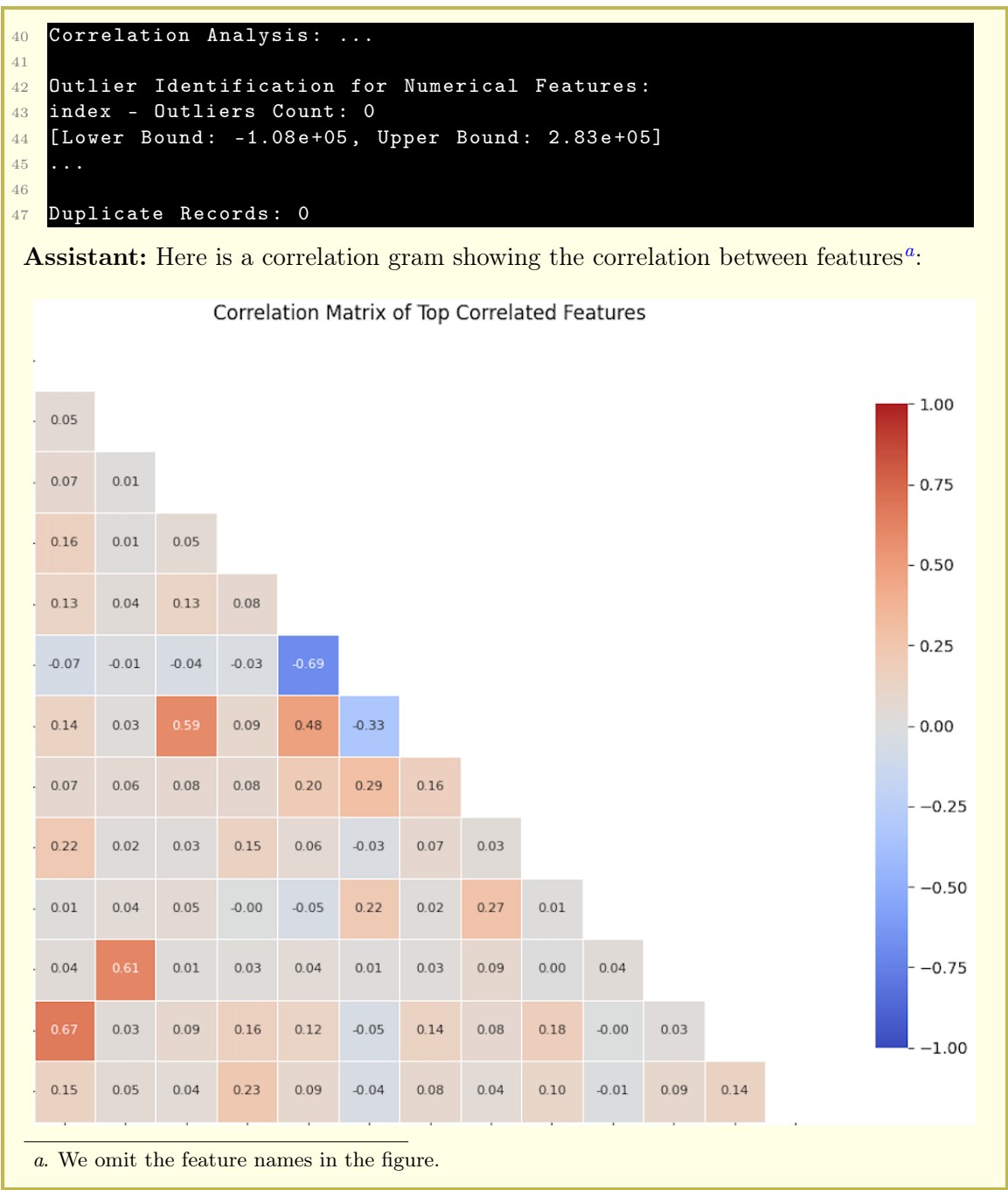

_a._ We omit the feature names in the figure.

---

**Assistant:** Would you like to generate descriptive statistics for the training dataset (lung_train.csv)?

---

**User:** Yes

**Assistant:** Tool `descriptive_statistics` completed successfully ✓

**Tool logs:**

```
1  Creating the descriptive statistics table...
2  Saving the summary table to:
3  /mnt/data-fourtb/Dropbox/Programming/wsl_repos/climb_versions/
       climb_cot/climb/wd/session_2025-01-08_14-33-29/lung_train.
       csv__descriptive_stats.csv
4  Identified numeric columns that should be considered categorical:
5  ['asbestos', 'lung_cancer', ...]
6  Creating plots for the data...
7  Normally distributed features:
8  []
9  Not normally distributed features:
10 ['age_at_recruitment', 'age_started_smoking', 'age_stopped_smoking', '
       body_mass_index', 'index', ...]
11 Plotted a bar plot for: 'sex'
12 Plotted a bar plot for: 'smoking_status'
13 Plotted a histogram and box plot for: 'age_at_recruitment'
14 ...
15 Plotting of 'years_since_stopped_smoking' took too long and was
       skipped.
```

**Tool output:**

```
1  Variable                    Summary
2  index                                            88896 (44130 -
       130874)
3  age_at_recruitment                                  59.1 (51.2
       - 63.7)
4  sex
5      Male                                             2483/4800
           (51.7)
6      Female                                           2317/4800
           (48.3)
7  ...
```

**Assistant:** The descriptive statistics have been generated successfully. Here is a summary of the key findings:

**Summary of Descriptive Statistics:**
Numerical Variables:

- age_at_recruitment: Median = 59.1 (Q1-Q3: 51.2 - 63.7)

- body_mass_index: Median = 27.1 (Q1-Q3: 24.5 - 30.4)

- ...

Categorical Variables:

- sex: Male = 51.7%, Female = 48.3%

- smoking_status: Previous = 75.1%, Current = 24.9%

- ...

**Key Figures:** Here are some important figures generated during the analysis:

- Distribution of age_at_recruitment:

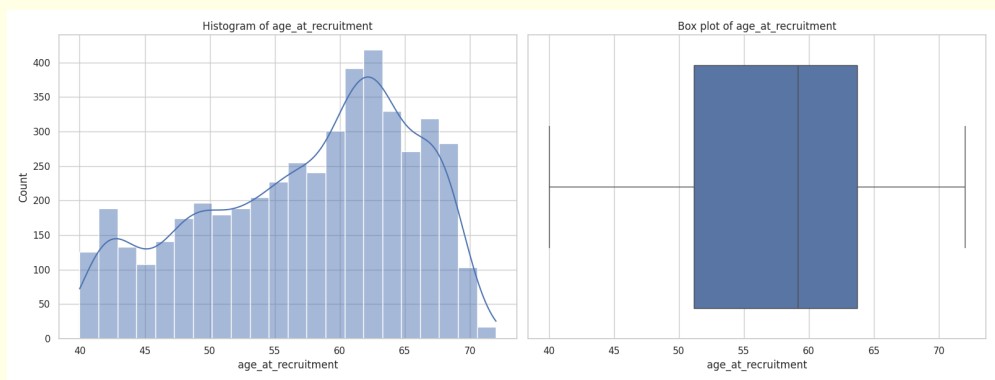

- Distribution of age_started_smoking:

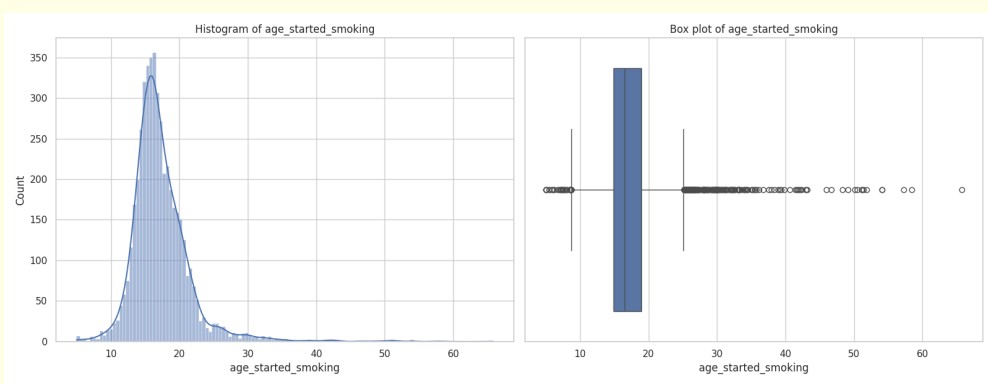

- Distribution of age_stopped_smoking:

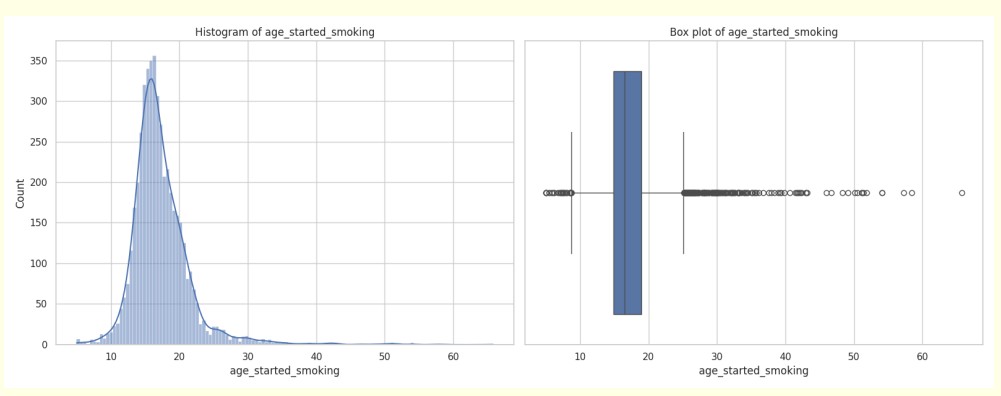

- Distribution of body_mass_index:

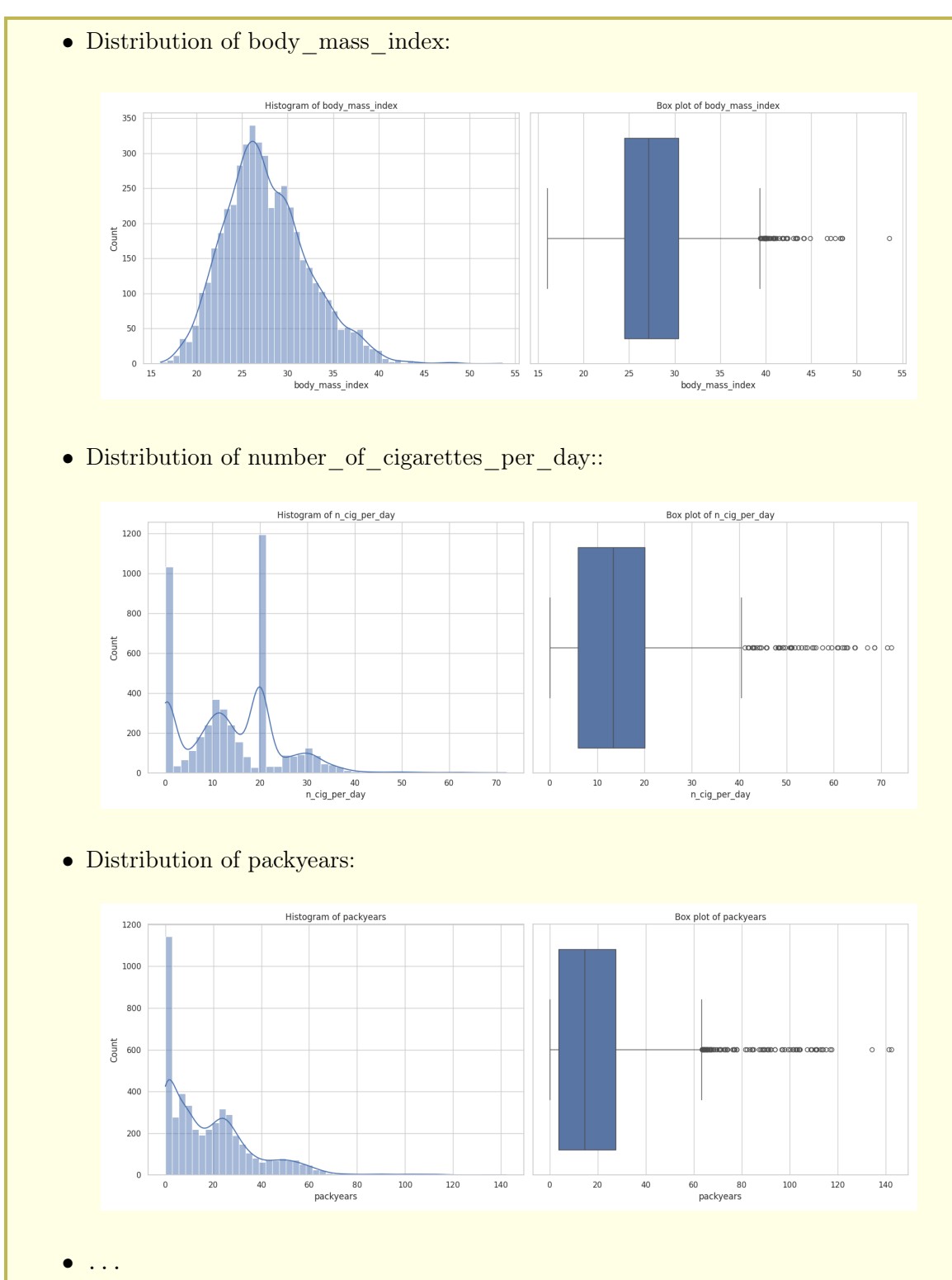

- Distribution of number_of_cigarettes_per_day::

- Distribution of packyears:

- . . .

**Assistant:** Would you like to see a Kaplan-Meier plot for the survival analysis?

**User:** Yes

**Assistant:** I will generate the Kaplan-Meier plot for the training dataset using the `lifelines` library.

**Generated code:**

```python
import pandas as pd
import matplotlib.pyplot as plt
from lifelines import KaplanMeierFitter

# Load the training dataset
df = pd.read_csv("lung_train.csv")

# Define the event and time columns
event_col = "lung_cancer"
time_col = "time_to_event_develop"

# Initialize the Kaplan-Meier fitter
kmf = KaplanMeierFitter()

# Fit the data
kmf.fit(durations=df[time_col], event_observed=df[event_col])

# Plot the Kaplan-Meier curve
plt.figure(figsize=(10, 6))
kmf.plot_survival_function()
plt.title("Kaplan-Meier Survival Curve")
plt.xlabel("Time to Event (days)")
plt.ylabel("Survival Probability")
plt.grid(True)

# Save the plot
plt.savefig("km_plot_training.png")
plt.close()
print("Kaplan-Meier plot for the training dataset has been saved as '
    km_plot_training.png'.")
```

**Code execution output:**
Code execution finished successfully ✓

```
Kaplan-Meier plot for the training dataset has been saved as '
    km_plot_training.png'.
```

**Assistant:** Here is the Kaplan-Meier plot for the training dataset:

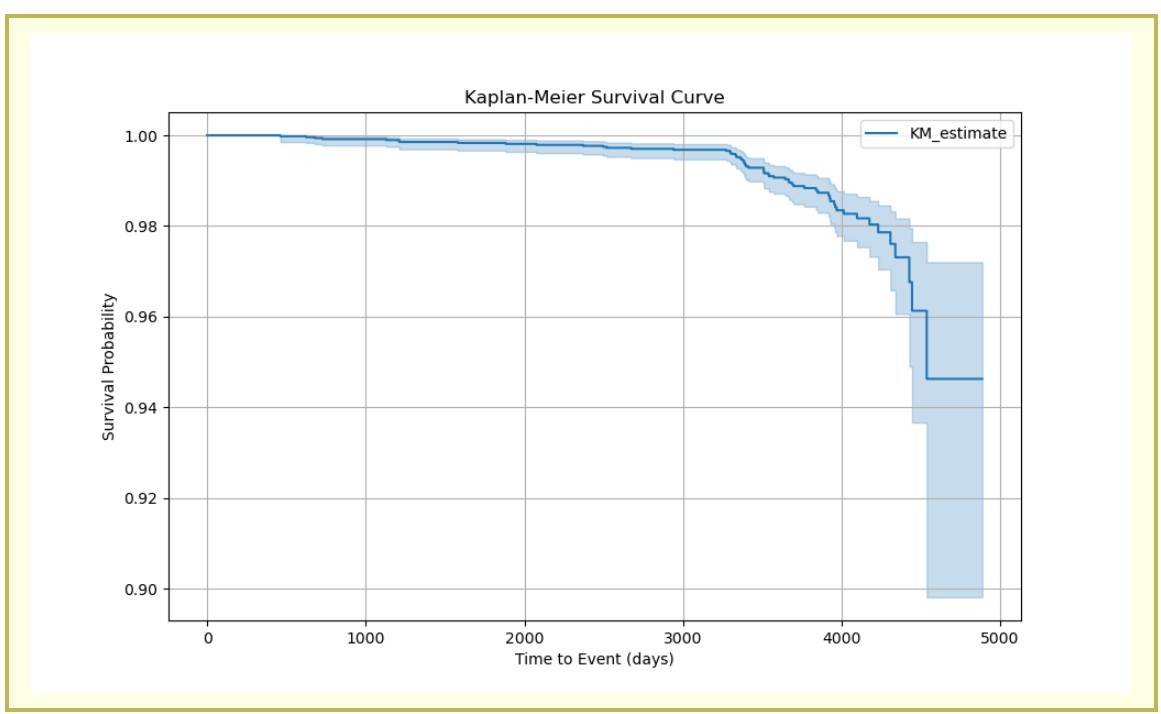

## D.2 Example: Data Aggregation

When input data contains multiple measurements, CliMB-DC will auto-check such issue and interact with users to address it.

---

**Summary of Progress to Date** The data file has been uploaded, and relevant background information has been provided to CliMB-DC.

. . .

---

Let's assess the suitability of your data for the task and whether the `AutoPrognosis` tools support it.

**Q1: Is the data suitable for the task?**

- The data is structured with each row representing a patient's information at different follow-up points.

- We need to predict the "Event_status" variable using the "Years_to_event" as the time variable, which indicates a survival analysis problem.

Potential issue: The data has multiple rows per patient, which might not be directly suitable for the `AutoPrognosis` tools that expect one row per patient.

**Q2: Does the AutoPrognosis set of tools support the task?**

- `AutoPrognosis` supports survival analysis through the `autoprognosis_survival_train_test` tool.

- This tool requires the target (event) variable and the time variable, which we have ("Event_status" and "Years_to_event").

---

**Conclusion**:

- The task can be performed using `AutoPrognosis`, but the data needs to be transformed to have one row per patient.

To transform the data, we can aggregate the follow-up points for each patient. This could involve taking the last follow-up point for each patient or aggregating information in a meaningful way.

Would you like to proceed with transforming the data to fit the task requirements?

**User:** I would like to take the last follow up point.

**Assistant:** Let's proceed with transforming the data to have one row per patient, taking the last follow-up point for each patient.

**Generated code:**

```python
import pandas as pd

# Load the data
file_path = "PBC_Data_corrected_v2.csv"
data = pd.read_csv(file_path)

# Sort the data by patient ID and follow-up time (assuming columns are
#       'patient_id' and 'Years_to_event')
# Replace 'patient_id' with the actual column name for patient
#       identification if different
data_sorted = data.sort_values(by=['patient_id', 'Years_to_event'])

# Take the last follow-up point for each patient
data_last_followup = data_sorted.drop_duplicates(subset=['patient_id'
    ], keep='last')

# Save the transformed data
output_file_path = "PBC_Data_last_followup.csv"
data_last_followup.to_csv(output_file_path, index=False)

print(f"Transformed data saved to {output_file_path}")
```

> **Code execution output:**
> Code execution failed ×
>
> ```
> KeyError: 'patient_id'
> ```

**Assistant:** It appears that the column name for patient identification is not "patient_id". Could you please provide the correct column name for patient identification?

**User:** Yes, that would be "id"

**Assistant:** Let's proceed with transforming the data using the correct column name "id" for patient identification.

**Generated code:**

```python
import pandas as pd

# Load the data
file_path = "PBC_Data_corrected_v2.csv"
data = pd.read_csv(file_path)

# Sort the data by patient ID and follow-up time
data_sorted = data.sort_values(by=['id', 'Years_to_event'])

# Take the last follow-up point for each patient
data_last_followup = data_sorted.drop_duplicates(subset=['id'], keep='last')

# Save the transformed data
output_file_path = "PBC_Data_last_followup.csv"
data_last_followup.to_csv(output_file_path, index=False)

print(f"Transformed data saved to {output_file_path}")
```

**Code execution output:**
Code execution finished successfully ✓

```
Transformed data saved to PBC_Data_last_followup.csv
```

**Assistant:** The data has been successfully transformed to have one row per patient, taking the last follow-up point for each patient. The transformed data has been saved as PBC_Data_last_followup.csv.
Based on the assessment:

- The data is now suitable for the task.

- The `AutoPrognosis` tools support survival analysis, which is the type of study you want to perform.

You can proceed with the survival analysis using the `autoprognosis_survival_train_test` tool.

### D.3 Example: Joining Multiple Files

Here, we show an example of CliMB-DC handling a complex case of data file merging. The Single Family Loan-Level Dataset (Freddie Mac (2025)) is split across 25 years, each year containing two files – requiring the stacking of the annual data followed by the merging of the origination and monthly data files by a key. In addition, here, the user has requested to use a subset of the years as the test set.

**Summary of Progress to Date** All the files have been uploaded, the files have been inspected and the missing headings added; the user has specified that they wish to use years 2018-2023 as the test set.

. . .

**Assistant:** To proceed, I'll help merge the provided datasets into single training and test files.

**Step I: Understanding the situation**
You have multiple files for both the training and test datasets:

1. Training set: Years prior to 2018.

2. Test set: Years 2018-2023.

**Questions:**

1. Do all the files within each group (training or test) contain the same kind of data and can be stacked directly, or do they differ and need to be merged by a unique key?

2. Is there a unique key or a combination of columns that can be used to merge the files, if needed? If you're unsure, I can assist in identifying one.

Let me know how you'd like to proceed!

---

**User:**

- The files for different years should be stacked into a single file.

- The origination and surv files should be joined on "LOAN SEQUENCE NUMBER".

- We will use years 2018-2023 (inclusive) as the test set.

---

**Assistant:** Understood. I will first stack the files for the training and test datasets separately, then merge the stacked "orig" and "surv" files using the "LOAN SEQUENCE NUMBER" column.

**Step II: Stacking Files** I will start by stacking the "orig" files for the training and test datasets.

Let me first stack the training "orig" files (years prior to 2018).
I'll now stack the "orig" files for years prior to 2018 (training dataset). These include the following files:

- `sample_orig_1999_fixed.txt`

- . . .

• `sample_orig_2017_fixed.txt`

**Generated code:**

```python
import pandas as pd
import glob

# List of training "orig" files (years prior to 2018)
training_orig_files = [
    "sample_orig_1999_fixed.txt", "sample_orig_2000_fixed.txt", "
        sample_orig_2001_fixed.txt",
    "sample_orig_2002_fixed.txt", "sample_orig_2003_fixed.txt", "
        sample_orig_2004_fixed.txt",
    "sample_orig_2005_fixed.txt", "sample_orig_2006_fixed.txt", "
        sample_orig_2007_fixed.txt",
    "sample_orig_2008_fixed.txt", "sample_orig_2009_fixed.txt", "
        sample_orig_2010_fixed.txt",
    "sample_orig_2011_fixed.txt", "sample_orig_2012_fixed.txt", "
        sample_orig_2013_fixed.txt",
    "sample_orig_2014_fixed.txt", "sample_orig_2015_fixed.txt", "
        sample_orig_2016_fixed.txt",
    "sample_orig_2017_fixed.txt"
]

# Initialize an empty list to hold dataframes
dataframes = []

# Read and append each file
for file in training_orig_files:
    df = pd.read_csv(file, delimiter=",")  # Assuming comma delimiter
    print(f"Loaded {file} with shape: {df.shape}")
    dataframes.append(df)

# Concatenate all dataframes
stacked_training_orig = pd.concat(dataframes, ignore_index=True)

# Save the stacked file
output_file = "training_orig_stacked.csv"
stacked_training_orig.to_csv(output_file, index=False)
print(f"Stacked training 'orig' files saved as {output_file} with
    shape: {stacked_training_orig.shape}")
```

**Code execution output:**
Code execution finished successfully ✓

```
Loaded sample_orig_1999_fixed.txt with shape: (50000, 32)
...
Loaded sample_orig_2017_fixed.txt with shape: (50000, 32)
Stacked training 'orig' files saved as training_orig_stacked.csv
    with shape: (950000, 32)
```

. . . The analogous stacking steps were then carried out for the origination files test set, and then for the monthly data files training and test sets. The system then moved on to the merging step. . . .

---

**Assistant: Next Step: Merging Files** I will now merge the stacked "orig" and "surv" files for both the training and test datasets using the "LOAN SEQUENCE NUMBER" column as the unique key.

Let me start with merging the training datasets: `training_orig_stacked.csv` and `training_surv_stacked.csv`.

Here is the code to merge the stacked training datasets (`training_orig_stacked.csv` and `training_surv_stacked.csv`) using the "LOAN SEQUENCE NUMBER" column:

**Generated code:**

```python
import pandas as pd

# Load the stacked training datasets
orig_data = pd.read_csv("training_orig_stacked.csv")
surv_data = pd.read_csv("training_surv_stacked.csv")

# Merge the datasets on "LOAN SEQUENCE NUMBER" using an inner join
merged_training_data = pd.merge(orig_data, surv_data, on="LOAN
    SEQUENCE NUMBER", how="inner")

# Save the merged dataset
output_file = "training_merged.csv"
merged_training_data.to_csv(output_file, index=False)

# Sense check: Number of rows lost and NaN values introduced
rows_lost = max(len(orig_data), len(surv_data)) - len(
    merged_training_data)
nan_values = merged_training_data.isna().sum().sum()

print(f"Merged training dataset saved as {output_file} with shape: {
    merged_training_data.shape}")
print(f"Number of rows lost in the merge: {rows_lost}")
print(f"Number of NaN values introduced in the merge: {nan_values}")
```

> **Code execution output:**
> Code execution finished successfully ✓
>
> ```
> Merged training dataset saved as training_merged.csv with shape:
>     (950000, 34)
> Number of rows lost in the merge: 0
> Number of NaN values introduced in the merge: 2915132
> ```
>
> The training dataset has been successfully merged into `training_merged.csv` with 950,000 rows and 34 columns. There were no rows lost in the merge, but 2,915,132 NaN values were introduced. This could be due to missing data in either the "orig" or "surv" files.
>
> Next, I will merge the stacked test datasets: `test_orig_stacked.csv` and `test_surv_stacked.csv` using the "LOAN SEQUENCE NUMBER" column. Let me proceed.

The analogous merging step was then performed for the test set files, finalizing the data preprocessing.

## Appendix E. Additional details and experiments

### E.1 Failure modes

Like all LLM-based systems, CliMB-DC can face failures, for instance:

- Ambiguous data structures: When dataset features have ambiguous semantics that even domain experts struggle to interpret, the Coordinator may make incorrect assumptions.

- Novel data-centric issues: For novel data issues not represented in our taxonomy, initial planning may be suboptimal until expert guidance is provided.

- Computational resource limitations: For extremely large datasets or complex transformations, tool execution time can become prohibitive.

- Potential hallucinations: For example, when tools are unavailable and code generation is needed.

**Mitigations:** To mitigate these challenges: (i) we log all intermediate states and transformations for post-hoc auditing, (ii) we maintain checkpointed states for possible recovery, (iii) backtracking logic is triggered automatically when transformations reduce data quality (e.g., size < threshold, missing columns), (iv) for known tools we provide the API structure to the LLM and allow the LLM to call them as tools, rather than producing hallucinated code.

### E.2 Computing Resources and Implementation Details

The CliMB-DC framework was implemented in Python 3.9, with the user interface (UI) built using Streamlit 1.40 (Snowflake Inc. (2024)). For all experiments, the LLM backbone used for both CliMB-DC and the baseline co-pilots was `gpt-4o-2024-05-13` to ensure a fair comparison of reasoning capabilities. All experiments were conducted on a workstation equipped with a 10-core Intel Core i9-10900K CPU, 64 GB of RAM, and a single NVIDIA GeForce RTX 3090 GPU with 24 GB of VRAM and CUDA version 12.4. The wall clock run time of each CliMB-DC run ranges from several minutes to no more than 2 hours, depending on the dataset size and complexity.

### E.3 API Details and Extensibility

CliMB-DC is designed with a modular and extensible architecture to facilitate community contributions and ensure the framework can evolve alongside advances in data-centric AI. The core design philosophy centers on a clear separation between the reasoning engine and the executable tools. This section details the API structure, focusing on how new data-centric tools can be seamlessly integrated into the framework.

**Engine.** The system's backbone is the 'Engine' (located in `src/climb/engine`), which orchestrates the entire workflow. It manages sessions, agent interactions with the LLM backend (e.g., OpenAI, Azure), and the overall state of the data analysis pipeline. The 'Engine' takes user specifications and the dataset as input, consults the coordinator agent to generate a plan, and directs the worker agent to execute tasks.

**Tools.** The worker agent's more complex actions are primarily performed through a library of 'Tools' (located in `src/climb/tool`). Each tool is a self-contained module that executes a predefined function, such as data imputation or quality assessment. Tools are wrapped in a `ToolBase` class and are executed asynchronously in a separate `ToolThread`. This design prevents the user interface from blocking during long-running data operations. A `ToolCommunicator` object handles logging and streams output from the tool back to the user, ensuring transparency and real-time feedback.

A key goal of CliMB-DC is to serve as a platform for the data-centric AI research community. To this end, we have designed a straightforward process for integrating new tools. A developer can add a new tool by following these steps:

1. **Define the Tool Function:** First, create a standard Python function that implements the tool's logic. This function must accept a `ToolCommunicator` instance as its first argument to handle all output (e.g., printing progress or returning results).

2. **Create a Tool Wrapper Class:** Next, define a new class that inherits from the `ToolBase` abstract class. This class acts as a wrapper, connecting the tool's logic to the CliMB-DC engine.

3. **Implement the Execution Method:** Inside the wrapper class, implement the `_execute` method. This method is responsible for calling the tool function using our `execute_tool` helper, which manages the multi-threaded execution.

4. **Provide a Specification:** Finally, define the tool's metadata by implementing the `name`, `description`, and `specification` properties. The `specification` is crucial; it's a JSON-like schema (following OpenAI's function-calling schema OpenAI (2023)) that describes the function's signature, including its parameters, types, and descriptions. This schema is exposed to the LLM-based agents, allowing them to understand how to call the tool and what arguments to provide.

The following code (Listing 1) demonstrates how to create a simple tool that calculates the maximum value of a specified column in a CSV file.

```python
# NB: data_file_path and target_column will need to be defined in
    specification
def get_col_max(tc: ToolCommunicator, data_file_path: str, target_column:
    str):
    """Example tool function to get the max value of a column within a
        dataframe"""

    # Instead of printing to console, we print via the ToolCommunicator
    tc.print(f"Getting max value in column {target_column} from {
        data_file_path}")

    df = pd.read_csv(data_file_path)

    output_str = "Column max: " + str(df[target_column].max())

    tc.set_returns(output_str)

```

```python
class ColumnMax(ToolBase):
    def _execute(self, \*\*kwargs: Any) -\> ToolReturnIter:
        # NOTE: In general tools work on a separate directory
        data_file_path = os.path.join(self.working_directory, kwargs["
            data_file_path"])

        thrd, out_stream = execute_tool(
            get_col_max, # Our tool function defined above
            data_file_path=data_file_path,
            target_column=kwargs["target_column"],
        )

        self.tool_thread = thrd
        return out_stream

    # Other properties we want to define for reasoning / action units.
    @property
    def name(self) -> str:
        return "column_max"

    @property
    def description(self) -> str:
        return "This returns the value of a column within a dataframe"

    # Specification defines the parameters for the underlying tool
    #     function
    @property
    def specification(self) -> Dict[str, Any]:
        return {
                "type": "function",
                "function": {
                    "name": self.name,
                    "description": self.description,
                    "parameters": {
                        "type": "object",
                        "properties": {
                            "data_file_path": {"type": "string", "
                                description": "Path to the data file."},
                            "target_column": {"type": "string", "
                                description": "Target column"},
                        },
                        "required": ["data_file_path", "target_column"],
                    },
                },
            }
```

Listing 1: Example of extending CliMB-DC with a new tool. This involves defining a core function (`get_col_max`) and a wrapper class (`ColumnMax`) that inherits from `ToolBase` and provides metadata for the LLM agent.

**Extending the project plan** The reasoning and planning capabilities of CliMB-DC are guided by a dynamic project plan, which is constructed by the coordinator agent as a sequence of modular steps called *episodes*. The system maintains a library of these predefined episodes,

each representing a specific data-centric or modeling task. The coordinator can select, reorder, and adapt episodes from this library to build a tailored workflow that addresses the user's goals and the specific challenges of the dataset. This modular design allows the framework's capabilities to be easily expanded by adding new episodes to the library.

Each episode is defined in a JSON-like format with several key fields that provide structured guidance to the multi-agent system:

- `episode_id`: A unique string that serves as the primary identifier for the episode.

- `episode_name`: An optional, human-readable title for the episode.

- `episode_details`: The primary set of instructions for the worker agent. This field contains a detailed natural language description of the tasks to be performed, including conditional logic, user interaction points, and the specific tools to be invoked.

- `coordinator_guidance`: Optional high-level strategic advice for the coordinator agent. This helps the coordinator determine the episode's relevance and optimal placement within the overall project plan. For instance, it might suggest when the episode is most useful or what prerequisites must be met.

- `worker_guidance`: Optional low-level implementation details for the worker agent. This can include tips on handling specific data formats, managing edge cases, or clarifying tool usage to ensure robust execution.

- `tools`: A list of string names corresponding to the tools from the tool registry required for the episode. This allows the system to verify that the necessary components are available.

Adding a new episode is as simple as defining a new dictionary entry in the episode library. The following minimal example (Listing 2) illustrates how to add a new episode for generating a basic statistical summary of the dataset.

```
1  {
2      "episode_id": "DATA_SUMMARY",
3      "episode_name": "Generate Data Summary",
4      "episode_details": """
5  - Use the 'data_summary' tool to generate a statistical summary of the
       dataset.
6  - Display this summary to the user for their review.
7  """,
8      "coordinator_guidance": """
9  This is a good initial exploratory step to help the user understand the
       basic
10 statistical properties of their data, like mean, median, and variance.
11 """,
12     "worker_guidance": """
13 The 'data_summary' tool takes the data file path as input and returns a
14 pandas DataFrame. Ensure the output is formatted clearly for the user.
15 """,
16     "tools": ["data_summary"],
17 }
```

Listing 2: A minimal example of a new episode definition for the episode library. This episode instructs the agent to generate and display a statistical summary of the data.

## E.4 Experiment in a Non-medical Domain

To further evaluate the robustness and generalizability of our framework, we extend our case studies beyond the healthcare domain. This next experiment investigates CliMB-DC's performance in a financial context, specifically mortgage default prediction, which introduces a distinct set of data-centric challenges. This scenario allows us to assess the co-pilot's capabilities across three critical dimensions:

- **Scalability and Performance:** The experiment uses a dataset significantly larger than our previous examples, testing the framework's ability to operate efficiently at scale.

- **Complex Data-Centric Workflow:** The problem requires processing and merging multiple raw data files and performing non-trivial, context-dependent feature engineering to construct the final analytical dataset. This tests the coordinator's multi-step reasoning and planning capabilities.

- **Domain Adaptability:** The co-pilot must interpret and reason about domain-specific terminology and rules unique to the financial industry to correctly define the prediction task, thereby testing its adaptability to novel domains with different expert knowledge nuances.

The task is a time-to-event (survival) analysis using the Freddie Mac Single-Family Loan-Level Dataset (SFLD), Freddie Mac (2025). This publicly available dataset contains information on approximately 54 million U.S. mortgages and is a standard benchmark for credit risk modeling. For this experiment, we use a representative sample of approximately 1 million loan records.

The dataset is provided in two separate sets of annual (or quarterly) files:

1. **Origination File:** Contains static, time-invariant features for each loan at the time of its creation, such as the borrower's credit score, the original loan amount (UPB), loan-to-value (LTV) ratio, interest rate, and property details.

2. **Performance File:** Provides a longitudinal record of each loan's performance, with monthly updates on its balance, delinquency status, and final termination status.

A key data-centric challenge in this task is to correctly formulate the survival analysis problem by merging these two data sources and engineering the outcome variables. This involves:

- **Linking Records:** Joining the origination data with the final performance record for each unique loan using the `LOAN SEQUENCE NUMBER` after having combined multiple annual data files.

- **Correctly Handling Per-column Missingness:** The dataset contains column-dependent non-standard missing value indicators (e.g. '9' in `PROPERTY VALUATION METHOD`, '00' in the last two digits of the `POSTAL CODE`, etc.), which makes appropriate handling of these values additionally challenging.

- **Defining the Event and Censoring Status:** Correctly mapping the `ZERO BALANCE CODE` in the performance data to determine the loan's outcome. A loan is considered to have experienced the event (default) if it terminates due to a Third Party Sale (Code 02), Short Sale or Charge Off (Code 03), or REO Disposition (Code 09) Freddie Mac (2025) and is considered right-censored otherwise.

- **Calculating Time-to-Event:** The survival time must be calculated as the number of months between the `FIRST PAYMENT DATE` and the `ZERO BALANCE EFFECTIVE DATE`.

In this experiment, we use 25 annual origination files (1999-2023) and 25 corresponding performance files (with some initial preprocessing: only keeping the final month's rows and keeping only the zero-balance related columns). The information necessary to define the event and time variables, as well as the different missing indicator definitions (as well as the dataset overview and the research question description) were given to CliMB-DC and all baselines.

This setup tests whether CliMB-DC can reason through a multi-step data preparation pipeline, and construct a valid, ML-ready dataset for a complex modeling task in a new domain. The results can be found in Table 9.

Table 9: Results on the Freddie Mac Single-Family Loan-Level Dataset (SFLD). Note that for Data-Interpreter, a single training file and a test file are required, hence file merging was pre-executed by human assistance, significantly simplifying the task.

| Method | Human Assistance | Results Valid | C-Index | Failure Modes | % runs tested |
|---|---|---|---|---|---|
| Data-Interpreter | Files already merged | ✗ | | Failed to execute project pipeline | 100% |
| OpenHands | - | ✗ | | Failed to combine data files | 40% |
| | | ✓ | 0.592 | (Successful) | 60% |
| **CliMB-DC** (No Coordinator & No Tools) | - | ✗ | | Failed to combine data files | 100% |
| **CliMB-DC** (No Coordinator & With Tools) | - | ✗ | | Failed to combine data files | 40% |
| | | | | Incomplete preprocessing leading to tool failure | 60% |
| **CliMB-DC** | - | ✗ | | Failed to combine data files | 40% |
| | | ✓ | **0.816** | (Successful) | 60% |

The results in Table 9 underscore the heightened complexity of this financial prediction task. The multi-step process of merging numerous files and performing context-dependent feature engineering proved challenging for all co-pilots. We find that for the `gpt-4o` backbone (as used in all experiments): Data-Interpreter failed entirely, while OpenHands and the

ablated versions of CliMB-DC struggled with the initial data aggregation step, leading to high failure rates. In contrast, the complete CliMB-DC framework successfully navigated the complex pipeline in the majority of runs, achieving a robust predictive performance (C-Index of 0.816). However, the 40% failure rate, primarily at the file combination stage, indicates that large-scale, multi-file data wrangling remains a challenge for LLM-based agents.

## Appendix F. Social impact

An important dimension to consider for Generative co-pilots like Climb-DC is their social impact and ethical considerations, especially in high-stakes domains like healthcare.

We follow Solaiman et al. (2023), which offers a comprehensive framework for evaluating the societal implications of generative AI systems.

CliMB-DC aligns strongly with the context-aware evaluation emphasized by Solaiman et al. (2023). In contrast to evaluations conducted solely at the base system level, CliMB-DC is explicitly designed for deployment in context—operating as a human-guided system that integrates expert feedback during data curation and modeling. This enables our system to account for domain-specific feedback, norms, constraints, and potential harms (e.g., label leakage, underrepresented populations).

Furthermore, CliMB-DC directly addresses several categories from the Solaiman et al. (2023) framework :

- Disparate Performance: Our multi-agent reasoning system helps identify and mitigate healthcare data issues that can lead to performance disparities across demographic groups, while enabling domain experts to evaluate disaggregated performance metrics.

- Privacy and Data Protection: Our system is able to execute tools locally, with no patient-level data shared with the LLM, thereby protecting privacy and data.

- Trustworthiness and Autonomy: By making data-centric challenges explicit and addressing them with established data-centric tools, we establish trustworthiness. Moreover, by incorporating human guidance, we ensure that transformations remain clinically relevant. Finally, all reasoning and tool selections made by CliMB-DC are logged and auditable. This supports interpretability and provides a concrete mechanism for building user trust.

- Overreliance on automation: By requiring expert feedback loops, our system mitigates the risks of automation bias and blind trust in LLM outputs. This is especially important in healthcare, where hallucinated transformations could have downstream clinical implications.

- Inequality and Marginalization: Our framework specifically highlights how data issues if not addressed by co-pilots can amplify healthcare disparities or be harmful (i.e. current model-centric approaches). Not addressing such challenges and applying such co-pilots in clinical settings can lead to inequality of healthcare outcomes due to the subsequent harms. Hence, our work highlights the importance of a data-centric lens to mitigate this.

