# OpenReview forum: "Towards Human-Guided, Data-Centric LLM Co-Pilots"
_DMLR — Accepted by DMLR_

### Review · Reviewer_tD3o · 2025-03-20

**Recommendation:** 2
**Confidence:** 3

**Summary Of Contributions:**

This paper is an attempt to tackle a gap existing between LLM-based agents and domain experts, which is the fact that domain experts are not AI/ML experts, often do not code or are not familiar with data science techniques. They do however, know the data they work with every day very well and are familiar with data processing issues that they probably deal with manually on a daily basis. This paper tries to bridge this gap by introducing a human-in-the-loop feedback to help stir the agent in a direction where the data processing is done correctly thanks to the domain expert analysis. In particular the authors showcase the improvements this brings to model predictions with data related to the healthcare industry.

**Strengths:**

This work has strengths that rely on its objective to be used by domain experts. Because of that objective, the authors have categorized very well the pain points an autonomous agent usually struggles with, in particular in the context of healthcare applications. These data-centric issues are reported in their specialized taxonomy which is helpful.

Furthermore, the UI of the copilot looks neat and well designed for supporting non-AI/ML experts. It seems to allow for smooth interaction through a chat interface that includes views of datasets and statistical analysis plots and various figures the agent can generate. This interaction is in my opinion the strongest point of this work and the authors should give more impact to this. There are not enough details on that interface.

The workflow is also a strong element of this work, in particular the fact that the coordinator can backtrack and plan a few steps ahead gives a good flexibility and robustness to the whole pipeline.

Finally, the tool support and its extensibility are good standard practices for scalable adoption and support of open-source software; I firmly support this initiative from the authors.

**Audience:**

Yes

**Claims And Evidence:**

- Taxonomy of Challenges: categorized well and their impacts correspond to real-world issues that are described correctly. However not all of them are treated, e.g. multiple files (merging and avoiding duplicates ...)
- Data-Centric Tools: the authors integrate a very good list of data science tools that are readily usable by the agent which helps the performance. It is not clear what is the contribution here. I believe more details on the implementation side would be needed. I believe it is not easy to implement a framework where an LLM can understand documentations and use tools and libraries appropriately. So if they did that corectly it is in fact a good contribution and I invite the authors to share that engineering effort.
- Human-in-the-Loop Alignment: the term "contextual alignment" is not well defined and this alignment is also not well studied or supported by experiments. It is debateable that human feedback is alignment, one could argue that it is simply giving the LLM better instructions from a human expert. Allowing this feedback loop in the pipeline is good, but it is not what most of the literature means by "alignment" in the context of LLMs. I would argue this word is more used for instruction fine tuning or post-training methods in general.
- Multi-agent Planning an Reasoning: it is clear in the paper that the agent is planning and reflecting on the human expert feedback. It is actually a great feature of the pipeline that the agent can backtrack and refactor its plan. However it is not really novel. And I am not convinced that this qualifies as "multi-agent", it would require more details to be determined.
- Empirical case Studies: they are very well conducted and showcase the performance of the agent. The illustrations also convey the high-level information about the episodes and is well supported by the results tables. This part is a clear strength of this work.

**Datasets And Benchmarks:**

N/A

**Extended Submissions:**

This work is an extension of [Climb: An ai-enabled partner for clinical predictive modeling](`https://github.com/vanderschaarlab/climb`). It builds on top of the previous ecosystem to add data-preprocessing tools. It is sufficiently different in terms of code contributions I believe, but this work should insist more on the core difference with its own previous version. In particular, can the ablation `CliMB-DC (No Coordinator & No Tools)` be considered as the previous version of `CliMB`? If so then I have no issues.

**Limitations:**

In general, there is a lack of novelty in this work. The authors state that the key novelty lies in enabling the method to backtrack and consult experts on errors. But there are a number of related works that already support such interactions. It would at least be good to see them as baselines in order for this work to claim superior performance.

Furthermore, the modality support seems limited to tabular data. The agent already can support doing some statistical analysis and plots for the user. But the agent needs the human to analyze these to progress. This could be done with the help of VLMs as well to not have to bother the expert on all visual analysis. Plus, some tasks might require actually classifying images.

**Requested Changes:**

### 1. Related Work
These related works should be at least mentioned, if not further discussed. Some might even be relevant baselines.
- Data-cleaning: `CleanAgent`, `LAMBDA`, `MatPlotAgent`, `WaitGPT`
- Agent copilots: `SEED: Domain-Specific Data Curation`, `AutoM3L`, `AutoML-Agent`, `HuggingGPT`, `MLCopilot`, `SELA`, `AutoKaggle`, `AgentK`

### 2. Experiments
There are only 3 experiments in this work. I appreciate they are very specialized and showcase the usefulness of the agent, but it would be interesting to have more experiments in different domains. In particular with varying degrees of experts and on varying task difficulties. This would clarify the performance difference between this work and baselines.

### 3. Multiple File sources problem
There is a mention of a specific problem in the taxonomy which is the multiple files problem. This appears to not be addressed in any of the experiments. This is often a difficulty for agents, i.e. to know what files to merge and how.

### 4. Clarity
There needs to be more clarity on the details or the backend and the pipeline.
- What LLMs are used and how do they differ? I.e. are there different LLMs between the coordinator and the worker? Are they different backends or are they specialized to certain tasks through fine tuning? A different system prompt does not qualify as a “multi-agent” setting.
- Why the LLM asks for human feedback and when? Is it always the case? What if the agent keeps going without asking for feedback? Is this behavior controlled and how?
- What is really in the plan of the coordinator? How is the coordinator prompted? What is included in its context?
- In Equation (4) it is stated that the plan at step t is conditioned on all sates from t=0 to t-1, but where are these on the right-hand side?
- In Equation (5) why is there a marginalization over all history of interactions? Does this mean in practice the coordinator sees all past human interventions to predict the next observation O^t?
- In M-Steps Lookahead Planning paragraph, what is meant by “episodes”? Is it used as a synonym for “steps”?
- How are the tools used? Though APIs or though code? Does the worker need to write code to use them? Can we see examples of that?
- How is the worker prompted?
- Can the agent see the visualizations or plots from the EDA or statistical analysis? Or are these only for the user’s consumption?
- The difference between S the state and H the history is not clear. Is the former included in the latter? In Figure (6) it seems that the new plan depends on H but it is formalized nowhere.
- At the end of the pipeline, what is the result for the user? Can the whole workspace be re-used or saved? Is all the code the worker writes saved somewhere so it can be inspected?
- The experiments show results averaged over 5 seeds, could there also be a standard deviation or standard error to see a dispersion?

### 5. Typos
Finally, here are some typos I believe:
- On page 3 in the first paragraph there is a whole sentence duplicated/paraphrased: “by integrating advanced data-centric … real-world settings”
- Equation (5) is the distribution of observation at step t conditioned on the state at step t-1 but the sentence above conditions the observation of the state at step t.
- Page 23 case study 3 “sculpting” seems an odd verb to use here, maybe “carved out”?

**Strengths And Weaknesses:**

### Strengths
The strengths of this work are that the authors are tackling an actual problem that is useful to users of LLM-based agents. By going the route of a co-pilot, they can rely on the knowledge of the expert and get around some difficulties that would be hard for an automated agent with no external feedback to avoid.

They have a strong list of issues agents usually face when preprocessing the data that can impact the outcome of a prediction. The taxonomy of such issues is a strength here as it helps planning accordingly. There is also a very good list of tools that are available to the agent to use (I am assuming through an API) which can make a difference in that setting. Finally, the UI looks well designed for the purpose of expert-agent interaction.

### Weaknesses
There are however, a number of weaknesses that need highlighting with the most important being novelty. This work does not introduce any new techniques or pipelines. It also misses to cite a number of other prior works (detailed below) that tackle a similar problem with similar solutions, i.e. bridging the gap between LLM agents and experts with copilot assistants. There are even some prior works that focus explicitly on data processing which in my opinion should at least be mentioned and better, be treated as baselines.

There is also a lack of clarity in section 4 and in general in the paper around the concept of multi-agent. It is not clear why the coordinator and the worker are multiple agents. Are they different LLMs? How do they differ? Are they fine tuned to specific tasks? If the only difference is in the system prompt this does not qualify as multiple agents. There needs to be in general more details about the backends here, i.e. LLMs used. There are also a number of approximations in the formalism.

Finally, the comparison with MCTS seems a little out of place. To the best of my knowledge, no related work on LLM agents is trying to explore or plan/reason with MCTS. In that space, it seems obvious that it would be a bad idea. So, I am not convinced this comparison brings anything to the discussion. But if there exists such related work, i.e. an LLM-based agent framework where planning and reasoning is done through MCTS, it would be good to cite and compare against it. If, however, other baselines used in that paper such as `DataInterpreter` and `OpenHands` are considered to be MCTS-base frameworks, then it should be stated more clearly and I would also disagree with that statement as there is possibility of self-reflection and dynamic re-planning for both frameworks. The difference with this work, again, relies in the human expert.

---

### Review · Reviewer_ixn3 · 2025-03-26

**Recommendation:** 3
**Confidence:** 2

**Summary Of Contributions:**

This paper introduces CliMB-DC, a novel framework for LLM-based machine learning co-pilots that specifically addresses the often-overlooked data-centric challenges inherent in real-world datasets, particularly in healthcare. Unlike existing co-pilots focusing mainly on model selection and tuning, CliMB-DC integrates state-of-the-art data-centric tools (for issues like missing data, leakage, data quality) with a sophisticated multi-agent reasoning system (a strategic Coordinator and an execution-focused Worker). It incorporates a human-in-the-loop approach, allowing non-technical domain experts to guide the data preparation and modeling process using natural language, ensuring transformations align with contextual knowledge.

**Strengths:**

See Strengths And Weaknesses

**Audience:**

Yes

**Broader Impact Concerns:**

The social impact and ethical considerations are worth more discussion. Some related works (such as [1]) may be helpful for the authors. It is recommended that the authors may provide more formal discussions with the guidance of these related works (such as [1]) if possible.

[1] Evaluating the Social Impact of Generative AI Systems in Systems and Society https://arxiv.org/abs/2306.05949

**Claims And Evidence:**

See Strengths And Weaknesses

**Datasets And Benchmarks:**

N/A

**Extended Submissions:**

N/A

**Limitations:**

See Strengths And Weaknesses

**Requested Changes:**

See Strengths And Weaknesses

Additional Comments:
1. The social impact and ethical considerations are worth more discussion. Some related works (such as [1]) may be helpful for the authors. It is recommended that the authors may provide more formal discussions with the guidance of these related works (such as [1]) if possible.

[1] Evaluating the Social Impact of Generative AI Systems in Systems and Society https://arxiv.org/abs/2306.05949

**Strengths And Weaknesses:**

Pros:
1. Directly tackles the crucial but often neglected data-centric aspects of the ML pipeline, which are major bottlenecks in real-world applications.
2. Effectively incorporates domain expertise through natural language interaction, crucial for navigating the complexities and constraints of real-world data (especially in high-stakes fields like healthcare).
3. Introduces a sophisticated Coordinator-Worker agent system with explicit reasoning steps (observation, backtracking, planning) designed for complex, adaptive workflow management.
4. Leverages and makes accessible state-of-the-art data-centric AI tools, bridging the gap between cutting-edge research and practical application for non-experts.
5. Provides a useful taxonomy of data-centric challenges (Table 1), offering a structured way to understand and address these issues in co-pilot development.


Cons:
1. While the multi-agent reasoning process (Coordinator with SO, BA, LP stages) is conceptually well-defined, the paper lacks detail on its concrete implementation. How exactly does the LLM execute these sophisticated reasoning steps (State Observation, Backtracking Assessment, Lookahead Planning)? Is it purely prompt-based? How robust is this reasoning to variations in LLM capabilities, prompt phrasing, or unexpected data states? The reliance on the LLM's reasoning fidelity for critical steps like planning and backtracking needs more rigorous validation or discussion of potential failure modes.
2. The case studies effectively demonstrate CliMB-DC's advantages on specific healthcare datasets with known data-centric challenges (leakage, aggregation, quality/drift). However, the scope is somewhat limited. How well would the framework perform on: (1) Much larger datasets? (2) Different types of data-centric problems not covered (e.g., complex feature interactions, high-dimensional sparse data, severe concept drift)? (3) Domains outside healthcare with distinct data characteristics and expert knowledge nuances? (4) Further evaluation across a wider range of scenarios would strengthen the claims of broad applicability.
3. The human-in-the-loop aspect is central. However, the paper could provide more detail on how human feedback was operationalized in the experiments, especially for ensuring consistency when comparing with baselines that also received "Human Assistance". How sensitive is CliMB-DC to the quality, timeliness, and ambiguity of user feedback? Exploring scenarios with suboptimal or conflicting expert input would be valuable.
4. The paper highlights the open-source nature and extensibility for new data-centric tools (Section 5.1), claiming "minimal effort" for integration. Providing more concrete details on the API, the actual process for registering a new tool, and the potential technical challenges involved would make this claim more convincing and useful for the community.
5. The results tables (4, 5, 6) effectively highlight the failure modes of baseline co-pilots. However, there's little discussion about the potential failure modes of CliMB-DC itself. Can the coordinator misinterpret user input, make poor planning decisions, fail to identify a necessary backtracking step, or hallucinate incorrect code/explanations? Acknowledging and analyzing potential limitations would provide a more balanced perspective.
6. Specifying the exact Large Language Model (e.g., GPT-4, Claude 3 Opus) used for the coordinator and worker agents would enhance reproducibility, as performance can be highly dependent on the underlying model.

---

### Review · Reviewer_Lu6t · 2025-07-10

**Recommendation:** 3
**Confidence:** 2

**Summary Of Contributions:**

The authors introduce CliMB-DC, a human-guided, data-centric framework for Large Language Model co-pilots. Their contributions include a multi-agent reasoning system with strategic coordination and specialized execution, which integrates state-of-the-art data-centric tools into an open-source, extensible architecture. They also provide a formalized taxonomy of data-centric challenges that are specifically relevant to LLM co-pilots. Furthermore, the authors implement a human-in-the-loop approach to ensure alignment with domain-specific constraints.

**Strengths:**

The paper addresses an important gap in the field by explicitly focusing on data-centric challenges that existing model-centric LLM co-pilots often overlook. By providing a detailed taxonomy, the authors deliver a structured scheme that will benefit future research and practical implementations. Their novel multi-agent approach enhances reasoning capabilities within LLM co-pilots, and the integration of human expertise ensures that the resulting data transformations are contextually aligned with domain-specific needs. The empirical studies provide evidence of real-world performance improvements which support the authors' claims. Additionally, the consideration of ethical implications through human-in-the-loop interactions further support the submission's strengths.

**Audience:**

Yes

**Broader Impact Concerns:**

Potential data protection issues should be explicitly considered and addressed, given the sensitive nature of the data likely to be used by this software. Ensuring compliance with data protection regulations and the secure handling of personal or sensitive data should be clearly stated and adequately integrated into the framework's safeguards.

**Claims And Evidence:**

This submission aligns with DMLR’s acceptance criteria: it fits the journal’s scope, is technically sound, and presents novel contributions. However, it falls short in empirical precision, reproducibility and ethical compliance. Addressing the need for stronger statistical analysis and clearly defining data protection safeguards will be essential to fully meet DMLR’s acceptance standards.

**Datasets And Benchmarks:**

While the paper and supplementary material describes the dataset in moderate detail and outlines their characteristics and tasks, they do not provide sufficient detail to support reproducibility. In particular, no dataset access URLs, licensing or hosting information or intended use documentation are included. The paper also does not specify any maintenance plan for these datasets or standardized splits for benchmarking.

**Extended Submissions:**

The submission is positioned as an extension of the previously published CliMB framework. The authors clearly cite and describe the earlier work, identifying CliMB-DC as an advancement through the addition of a novel multi-agent reasoning system, integration of data-centric tools, and human-in-the-loop mechanisms. However, the paper does not provide an explicit and systematic comparison specifying which sections, methods, experiments or analyses are newly introduced and which are carried over from CliMB.

**Limitations:**

The evaluation conducted exclusively on healthcare datasets restricts the demonstrated generalizability to other domains. Additionally, the statistical analysis, based on a limited number of repetitions, does not robustly support strong conclusions. The lack of a detailed failure mode analysis (the supplementary data includes only a few examples) further diminishes the impact and clarity of the presented empirical results.

**Requested Changes:**

It would be beneficial to provide empirical validation or explicit justification for the claimed applicability beyond healthcare datasets. Increasing statistical interpretability, either through more repetitions or a clearer acknowledgment of the limitations of the current approach (averaging over five runs), is recommended. Simplifying visualizations, particularly Figures 8–10, would enhance readability. Additionally, Tables 4–6 are difficult to read due to too many unexplained features. Including detailed hardware or computational resource requirements, alongside a description of the user interface, would better inform potential users.

**Strengths And Weaknesses:**

**Strengths:**
- Relevant and clearly defined research problem.
- Comprehensive taxonomy of data-centric challenges.
- Methodological advance through a multi-agent reasoning approach.
- Effective human-in-the-loop integration.
- Demonstrable empirical improvements on healthcare datasets.

**Weaknesses:**
- Lack of empirical validation for applicability beyond healthcare.
- Limited statistical interpretability due to averaging over only five runs.
- Complex and hard-to-interpret visualizations and tables.
- Redundant content in introductory chapters.
- Some of the mathematical formalism appears redundant, as the accompanying text already provides clear descriptions; the formulas do not substantially contribute to understanding.